# High-resolution proteomics unveils salivary gland disruption and saliva-hemolymph protein exchange in *Plasmodium*-infected mosquitoes

*Plasmodium* sporozoites, the infective stage of malaria, must invade the mosquito salivary glands (SGs) before being transmitted to a vertebrate host. However, the physiological and biochemical effects of this invasion remain largely unexplored. We examined the impact of *Plasmodium* infection on *Anopheles gambiae* salivary glands using high-resolution proteomics, gene expression, and morphological analysis. The data reveal differential expression of various proteins, including the enrichment of hemolymph-derived humoral proteins in infected salivary glands. These proteins diffuse into the SGs due to structural damage caused by the sporozoites during invasion, while saliva proteins diffuse out into the circulation. Moreover, proteomic analysis of saliva from *P. berghei*– or *P. falciparum*–infected mosquitoes revealed changes in composition, with a pronounced reduction of immune proteins relative to uninfected mosquitoes. This reduction is likely due to the association of these proteins with the surface of sporozoites and/or changes in the saliva's physical properties within the invaded salivary secretory cavities. The saliva protein profiles from mosquitoes infected with both *Plasmodium* species are remarkably similar, suggesting a conserved interaction between sporozoites and salivary glands. Our results provide a foundation for understanding the molecular interactions between *Plasmodium* sporozoites and mosquito salivary glands.

Malaria, a major mosquito-borne disease, accounted for 249 million cases and 606,000 fatalities in 2022[1]. The disease is caused by *Plasmodium* parasites and is transmitted to humans through the bite of an infected *Anopheles* mosquito. To curtail the spread of malaria, different countries have effectively adopted vector control programs by using insecticides. However, the emergence of insecticide-resistant mosquitoes has impeded elimination efforts, leading to the rebound of the disease in some countries[2]. Therefore, alternative strategies to thwart parasite transmission by infected mosquitoes are needed.

The development of *Plasmodium* parasites in mosquitoes is a lengthy, complex process involving interactions with different mosquito tissues. Infection begins when a mosquito ingests gametocytes while feeding on an infected host. Within the first hour, fertilization occurs in the mosquito midgut, and 20 hours later, ookinetes invade the midgut epithelium and differentiate into oocysts. Each oocyst undergoes sporogony, producing thousands of sporozoites. Around 9 to 12 days post-midgut invasion, sporozoites emerge from the oocyst and journey through the mosquito hemolymph to the salivary glands (SGs), which they invade[3–5]. Sporozoites can reside within the SGs for

✉e-mail: thiagoluiz.alvessesilva@nih.gov; joel.vega-rodriguez@nih.gov

the entire mosquito lifespan, entering a state of translational repression in which sporozoites are believed to decrease their metabolic activity until they are delivered into the skin of a new host when they promptly reverse the translational repression[6–8]. Notably, SG sporozoites metabolize sugars and sustain active mitochondria, indicating that a basic level of metabolic activity is vital[9,10]. Much of sporozoite biology within SGs, including the mechanisms by which the sporozoite obtains nutrients to fulfill its specific metabolic requirements, remains unknown. A better understanding of sporozoite-SG interactions can guide the development of strategies to interrupt parasite transmission.

During mosquito probing, sporozoites are delivered into the host's skin along with mosquito saliva, which contains active molecules that interfere with host hemostasis and the immune response, including complement activity, inflammation, and white blood cell responses[11]. Transgenic mosquitoes lacking SGs display increased biting frequency and probing time, yet the volume of blood ingested remains unchanged, highlighting the importance of saliva in regulating feeding behavior[12,13]. Conversely, saliva proteins also influence parasite activity and, thereby, infectivity. The saliva protein gamma interferon inducible lysosomal thiol reductase-like protein (mosGILT) associates with the sporozoite surface, reducing sporozoite motility[14]. In contrast, the saliva protein sporozoite associated mosquito saliva protein 1 (SAMSP1) binds to the sporozoite surface, increasing gliding and cell traversal activity[15]. Interestingly, sporozoite infection of the SGs notably changes the saliva composition and decreases the activity of certain saliva enzymes through an unknown mechanism[16–18]. Some reports highlight that mosquitoes infected with sporozoites exhibit longer probing times and a greater propensity for multiple blood feedings, indicating modified feeding behavior[16,19–22]. However, other reports found no change in the probing time between infected and uninfected mosquitoes[5,23]. These discrepancies might result from the parasite-mosquito combination used for each study or time-sensitive changes of the SGs during infection[24]. Clarifying the impact of sporozoites on saliva composition is crucial for unraveling the dynamics behind host-pathogen interactions.

The SGs act as a critical bottleneck for sporozoite transmission – of the thousands of parasites that invade the SGs, only a small fraction is transmitted[5], making this stage an attractive target for transmission-blocking strategies. However, a comprehensive understanding of how sporozoites affect the protein composition in the SGs remains incomplete. Previous proteome studies have been limited by the identification of a small number of proteins and the use of total salivary gland homogenate rather than saliva[17,18]. Here, we performed high-resolution proteomics of SGs from *Plasmodium berghei*-infected or uninfected *Anopheles gambiae* mosquitoes to uncover the impact of sporozoites on SG protein expression. We also investigated the effect of *P. berghei* and *Plasmodium falciparum* sporozoites on saliva composition. We found that SGs are physically disrupted by parasites, leading to substantial exchange of proteins with the hemolymph. Our findings not only expand the list of previously identified proteins but also offer new insights into the structural and cellular biology of the mosquito SG during sporozoite infection and the potential effect on parasite transmission.

## Results

### The proteome of uninfected and *P. berghei*-infected *An. gambiae* SGs

To understand how *Plasmodium* sporozoites impact the protein composition of mosquito SGs, we performed high-resolution proteomics of *P. berghei*-infected SGs from *An. gambiae* mosquitoes 21 days post-infection, when salivary glands are fully invaded, and a time-point commonly used for sporozoite transmission studies (Fig. 1 and Supplementary Fig. 1). As a control, uninfected mosquitoes were blood-fed on an uninfected mouse at the same time as the infected group. We identified 3497 proteins comprising mosquito or parasite

proteins, each containing at least three unique peptides. Hence, 2816 *An. gambiae* proteins were identified in the uninfected SGS (uSGs) (average detection of 2577.67, $n = 3$), and 2842 proteins in *P. berghei*-infected SGs (iSGs) (average of 2647.33, $n = 3$) (Fig. 1a and Supplementary Data 1). Additionally, we detected 277 *P. berghei* sporozoite proteins (Supplementary Data 2).

To compare the composition of different functional groups between uSGs and iSGs, we used the relative abundance of protein families[25,26]. Using SignalP 5.0[27], we observed that proteins containing a signal peptide constituted 67% and 63% of the total protein mass in uninfected and infected samples, respectively (Fig. 1b). Functional annotation based on gene ontology revealed that the category of secreted proteins, which includes the classical saliva proteins and excludes transmembrane proteins or intra-cellular proteins, accounted for 66% of the total protein mass in uSGs and 62% of the total mass in the iSGs (Fig. 1c, d, and Supplementary Data 3). In contrast, protein synthesis, the second largest category, constituted approximately 9% of both groups (Fig. 1c, d and Supplementary Data 3). We observed a significant increase in the relative mass of 13 functional families, where lipid transport (5 proteins) and iron transport (4 proteins) had the most salient ratio increases of three- and two-fold, respectively (Fig. 1e, f).

### *Plasmodium* infection induces differential expression of salivary gland proteins

Principal component analysis (PCA) revealed that uSGs and iSGs formed two distinct clusters (Fig. 2a), showing that salivary gland invasion by *P. berghei* sporozoites impacts protein expression. A Venn diagram revealed that 31 proteins were exclusive to uSGs, 57 were exclusive to iSGs, considering proteins detected in at least two out of three replicates, whereas 2591 proteins were commonly detected in both uSGs and iSGs, considering proteins detected in at least four out of six samples, including uSGs and iSGs (Fig. 2b and Supplementary Data 4–6). Importantly, in this analysis, we considered only proteins identified with at least three unique peptides for stringency. Proteins exclusively detected in one condition were deemed differentially expressed, while commonly found proteins were separately analyzed for differential expression. A volcano plot of commonly found proteins revealed 24 upregulated proteins and seven downregulated (Fig. 2c), distributed across 13 functional classes (Fig. 2d). Remarkably, in the list of the classical saliva proteins, only three were differentially expressed: poly(U)-specific endoribonuclease (EndoU) was upregulated (Fig. 2d and Supplementary Data 7), and D7r3 and a 5' ectonucleotidase (5'-NTD) were downregulated 57% and 33%, respectively (Fig. 2e). The remaining saliva proteins had a consistent, but not significant, trend for lower expression (Fig. 2e). D7r3 binds tightly to histamine, a biogenic amine that promotes itching and is an agonist for platelet activation[28,29]. The 5'-NTD decreases hemostasis by converting AMP to adenosine, which antagonizes platelet aggregation by binding to the A2A receptor[30]. Adenosine promotes vasodilation[31] and immunosuppression[32]. *P. berghei* infection significantly upregulated EndoU (9.8-fold) (Fig. 2d). Although EndoU function in *An. gambiae* is unknown, in *Drosophila*, it plays pleiotropic roles, such as regulating muscle function and lipid metabolism[33,34].

Differentially expressed proteins in *P. berghei* iSGs, are involved in energy metabolism including substrate acquisition for oxidative metabolism, mitochondrial adaptations, and iron transport (Fig. 2d, e). Two alpha-amylases, which hydrolyze starch and glycogen into low-molecular-weight carbohydrates, were upregulated in iSGs (AGAP012401-PA, Fold Change (FC)=3.8; AGAP012400-PA, detected exclusively in iSGs), suggesting that these enzymes could support sporozoite survival by enhancing mosquito sugar feeding (Fig. 2d and Supplementary Data 7). Lipophorin (AGAP001826-PA), a key lipid transport protein, was enriched threefold in iSGs (Fig. 2d and Supplementary Data 7). Lipophorin is required for oocyst growth and

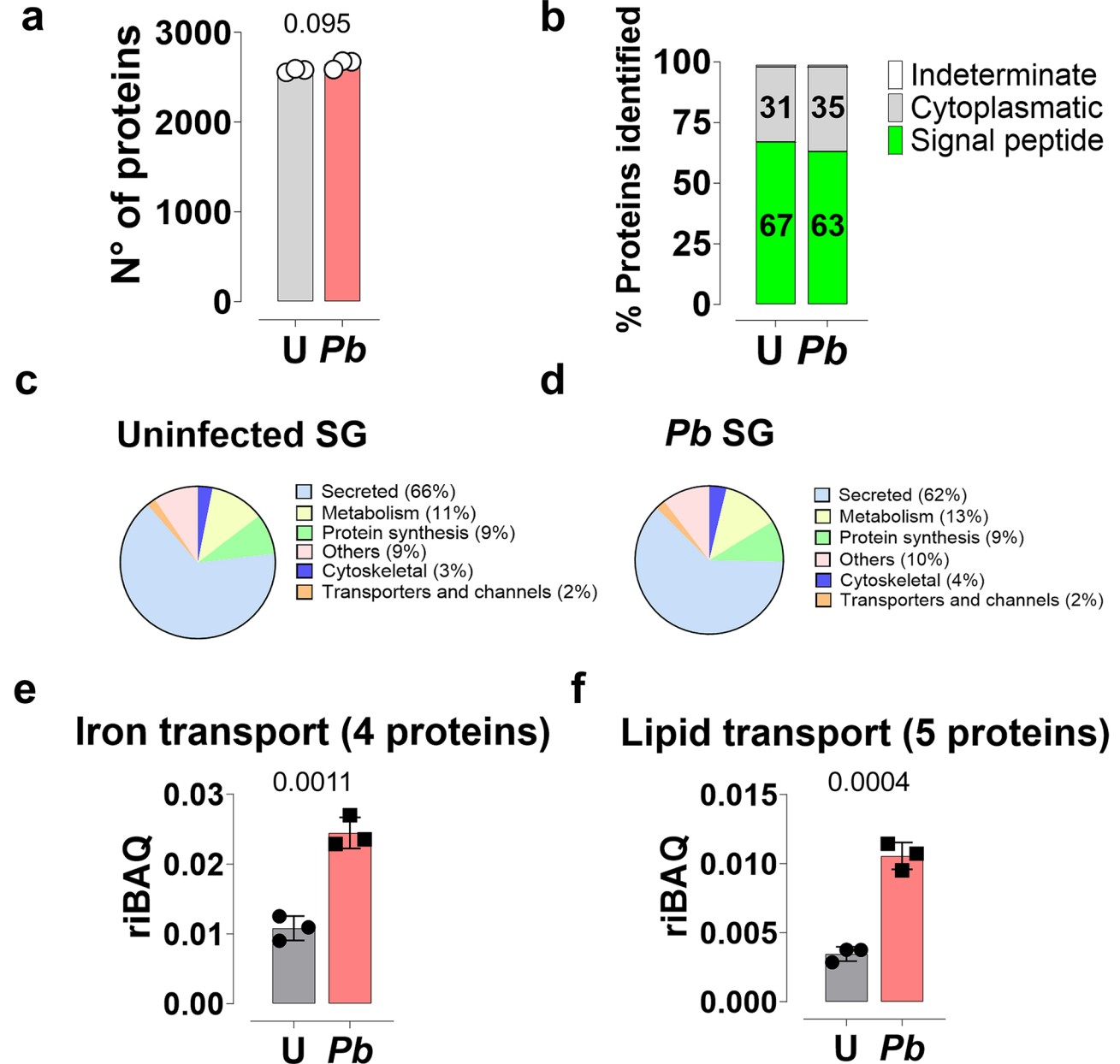

Fig. 1 | **Proteomic analysis of uSGs (U) and *P. berghei*-iSGs (*Pb*). a** Total proteins identified using MaxLFQ in uSGs and iSGs. Analyses were conducted on three independent pools (*N* = 3), each containing 25 pairs of SGs. The data are represented as mean ± standard deviation of the mean (SD). No significant difference was found using a two-sided *t*-test (*P* = 0095). **b** The classification of proteins predicted by SignalP 5.0 is shown as a percentage relative to iBAQ. Numbers inside the bar represent the percentage of each category. Relative protein abundance of functional classes in the salivary gland proteome of uninfected (**c**) and infected (**d**) mosquitoes, expressed as a percentage based on iBAQ values. Relative abundance of iron (**e**) and lipid (**f**) transport proteins in uSGs and iSGs. The number of proteins identified in each category is indicated in parentheses. For **e**, **f**, the data represent the mean values from three independent experiments ± SD. Statistical significance was determined using a two-sided Student's t-test ($P_{iron\ transport}$ = 0.0011, $P_{lipid\ transport}$ = 0.0004). Source data are provided as a Source Data file.

sporogony, and its depletion reduces sporozoite infectivity[10,35]. Immunofluorescence assays (IFA) and western blot confirmed higher lipophorin levels in iSGs, suggesting that sporozoites may have access to an abundant supply of lipids inside the SGs (Fig. 5b and Supplementary Fig. 2). Iron transport proteins, such as transferrin and ferritin, were enriched, possibly reflecting increased metabolic demand or nutritional immunity[36,37] (Figs. 1e, 2d).

The immune response in iSGs involves the upregulation of the prophenoloxidase pathway (PPOp) proteins, including those involved in pathogen recognition, signal amplification, and effector response (Supplementary Fig. 3C). Leucine-Rich Repeat Immune Protein 1 (LRIM1), a protein crucial for pathogen recognition, forms a complex with APL1C and TEP1 that activates the PPOp via Clip-domain serine proteases (CLIPs)[38,39]. LRIM1 was detected exclusively in iSGs (Supplementary Data 6), whereas TEP1 showed a non-significant two-fold increase. Among nine CLIPs identified, CLIPB4, an effector serine protease that leads to ookinete melanization[40], was upregulated 12-fold in iSGs (Fig. 2d, and Supplementary Data 6 and 7). Two PPOp negative regulators, CLIPA14 and CLIPA7[40,41] were up-regulated, with CLIPA14 accumulating in iSG cavities and on sporozoite surfaces (Supplementary Fig. 3a), while the expression in the hemolymph remained unchanged (Supplementary Fig. 3b). Four phenoloxidases

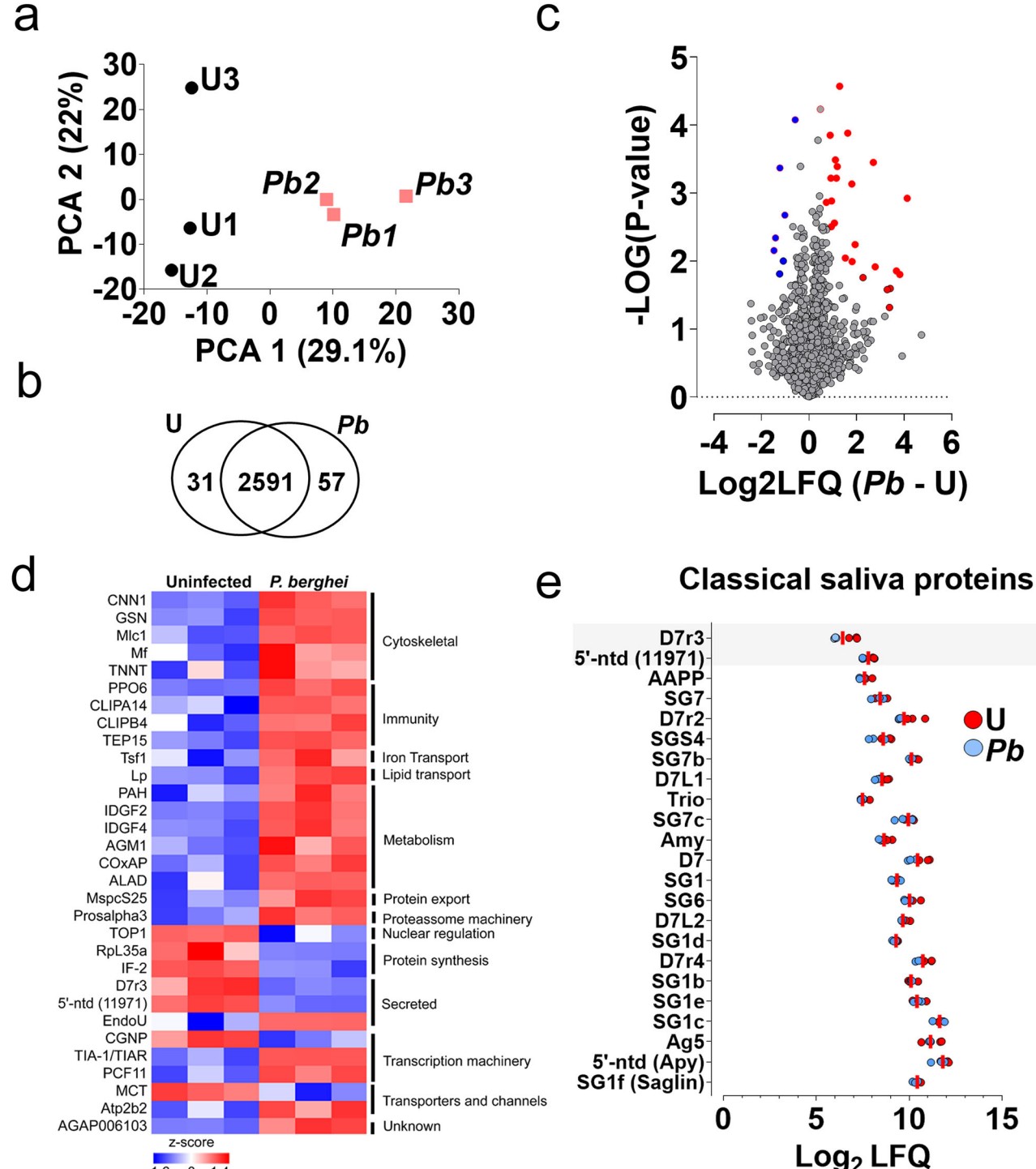

Fig. 2 | **Differential protein expression in uninfected (U) and *P. berghei*-iSGs (*Pb*). a** Principal component analysis (PCA) of normalized protein abundances based on MaxLFQ values for uSGs and *Pb* iSGs. Colors represent different experimental conditions, and labels identify individual samples. **b** Venn diagram showing the total number of proteins identified in independent samples. ($n = 3$). **c** Volcano plot of differential protein expression between uSGs and iSGs. The X-axis represents log₂ fold changes, while the y-axis shows significance ($-\log_{10}$ p-value). Proteins downregulated in iSGs are marked in blue, and proteins upregulated are marked in red. Only proteins detected in both conditions are included. Differential expression was tested with a two-sample Student's *t*-test with permutation-based

FDR control ($q \leq 0.05$) using a SAM-style variance parameter ($s_0 = 0.1$). **d** Heatmap of 21 differentially expressed proteins between uSGs and *Pb* iSGs samples. Protein expression values were Z-transformed and normalized across conditions. Rows (individual proteins) are ordered by function and clustered using Euclidean distance. **e** Dot plot showing the abundance of classical saliva proteins in uSGs and iSGs. Each dot represents the Log₂ LFQ values for individual proteins in uSGs (red) and iSGs (blue). Proteins are listed on the vertical axis, and Log₂ LFQ values on the horizontal axis indicate relative abundance. The shadowed areas indicate abundant proteins differentially expressed. Source data are provided as a Source Data file.

(POs) were detected, with PPO6 upregulated 2.4-fold in iSGs (Fig. 2d) and strongly associated with sporozoite surfaces (Supplementary Fig. 3d). PPO9, known for mediating late-phase immunity against *P. berghei* oocysts, was detected solely in the uSGs (Supplementary Data 5)[42].

Despite the upregulation of PPOp components, sporozoite melanization has never been observed within SGs. Sporozoites evade melanization in the hemolymph through glutaminyl cyclase-mediated modifications[43]. However, glutaminyl cyclase knockout sporozoites were not melanized inside the SGs, suggesting an efficient mechanism to prevent melanization. Salivary peroxidase may inhibit melanization by depleting catecholamines, or D7 proteins abundant in saliva may sequester dopamine, preventing melanization within the glands[29,44-47]. TEP15 and gelsolin were also upregulated, indicating roles in immunity and clotting (Fig. 2).

### Accumulation of upregulated proteins in the iSGs

To validate our results, we performed IFA on uSGs and iSGs. We confirmed stronger TEP15, transferrin-1, and PPO6 staining in the iSGs compared to uSGs (Fig. 3a–c). The expression of the anopheline antiplatelet protein (AAPP), a classical saliva protein expressed in the distal region of the lateral lobes, remained unchanged, corroborating our proteome data (Fig. 3d). In agreement with increased PPO levels in iSGs, we consistently observed discrete melanization spots on the surface or within the iSGs, which were prevalent in infected glands (Fig. 3e and f). Interestingly, melanization was limited to isolated spots, and widespread melanization did not occur in any iSGs, suggesting that sporozoites either inhibit the process directly or exploit a mosquito mechanism that prevents melanization in the SGs.

### Protein accumulation in iSGs occurs independently of transcriptional activity

In the iSG proteome, we observed the enrichment of proteins typically produced in mosquito tissues other than the SGs. Notable examples include lipophorin and CLIPA14, which are chiefly expressed in the fat body[48,49]; transferrin 1, highly expressed in antimicrobial hemocytes[49]; and PPO6, expressed in hemocytes[49,50]. We hypothesized that these proteins could either be synthesized in the infected iSGs or be imported from the hemolymph. To determine whether *P. berghei* infection induces the transcription of these genes in the SGs, we performed qPCR on uSGs and iSGs 20 days post-infection. As a positive control, we selected five saliva genes known to be highly expressed in the SGs: three genes that were not differentially expressed in the proteome (antigen 5, apyrase, and AAPP) and two genes that were downregulated (5' ectonucleotidase and D7r3). No significant difference in transcription was observed for these genes between uSGs and iSGs, except for AAPP and D7r3, which were strongly upregulated in iSGs (Fig. 4a). The upregulation of D7r3 transcription could be a response to the reduced protein levels upon sporozoite infection, whereas the strong upregulation of AAPP transcription is puzzling and requires further study to understand its regulatory mechanism upon sporozoite invasion. Then, we tested the transcriptional activity of four genes known to be produced outside the SGs. In iSGs, CLIPA14 and PPO6 mRNA expression were downregulated, while transferrin and lipophorin levels remained unchanged (Fig. 4b). Note that the transcript levels for these genes are extremely low. Presumably, these mRNAs originate from other cells that remain attached to the SGs after dissection, like fat body and hemocytes.

To pinpoint whether these genes were expressed in the SGs or any tissue attached to them, we performed in situ hybridization of uSGs and iSGs using specific probes for lipophorin, PPO6, and apyrase. In uSGs, apyrase expression was localized to the distal portion of the central lobe and a distinct spot in the proximal region of the lateral lobes (Fig. 4c and Supplementary Fig. 4). In iSGs, apyrase showed a similar pattern, except in the proximal region of the lateral lobes,

which exhibited strong transcriptional activity. This suggests that SG infection can alter the transcriptional pattern of saliva proteins, which are traditionally believed to be restricted to specific SG sites. Notably, a signal for apyrase's mRNA was detected within the salivary cavity, suggesting a significant extracellular localization (Fig. 4c and Supplementary Fig. 4). Mosquito saliva is known to contain extracellular RNAs that can be enclosed in extracellular vesicles[51,52]. Whether this applies to apyrase RNA and plays a role in salivary gland biology remains to be explored. Lipophorin and PPO6 expression were confined to their expected tissues: fat body cells and hemocytes attached to infected salivary glands, respectively (Fig. 4c–g). These results support that lipophorin, PPO6, and presumably other humoral proteins enriched in iSGs originate from external tissues. Henceforth, we will refer to these proteins as enriched or depleted instead of up- or downregulated.

### Sporozoite invasion compromises the SG epithelial integrity

In the *P. berghei* iSG proteome, many enriched proteins matched those circulating in the hemolymph in high concentrations. Since it is unlikely that SGs produce some of these proteins (e.g., lipophorin and PPO6), we hypothesized that hemolymph proteins might enter the SGs due to structural damage inflicted by sporozoites during salivary gland invasion[24,53]. Consequently, we also expected to find saliva proteins in the hemolymph of infected mosquitoes. To gain insight into the hemolymph composition of infected mosquitoes, we performed high-resolution proteomics on *An. gambiae* hemolymph 19 days post-infection, when many sporozoites had invaded the SGs (Supplementary Data 8). This analysis confirmed that several proteins abundantly present in the infected hemolymph coincided with those enriched in the iSGs (Fig. 5a, Supplementary Figs. 5a, b, and Supplementary Data 6 and 7). Notably, we detected abundant saliva proteins, such as apyrase and antigen 5, among those found exclusively in the *P. berghei*-infected hemolymph (Supplementary Fig. 5c, and Supplementary Data 8). Western blot analysis using antibodies against lipophorin (a hemolymph marker) confirmed that lipophorin was enriched in the iSGs but not in the uSGs (Fig. 5b). In contrast, apyrase (a saliva marker) was detected in the hemolymph of infected mosquitoes 21 days post-infection but not at 9 days post-infection when the SGs had not yet been invaded (Fig. 5c).

To investigate whether sporozoite invasion compromised the epithelial integrity of iSGs, we measured the diffusion of fluorescent dextran from the hemolymph into the SGs after intrathoracic injection in uninfected and infected mosquitoes 21 days post-infection. We found that 90% of the iSGs were permeated by dextran, compared to only 10% of the uSGs (Fig. 5d). However, injection of dextran at 9 days post-infection, before SG invasion, showed that only 10% were permeated in both groups, infected and uninfected SGs (Supplementary Fig. 9a). These results confirm that sporozoite-induced epithelial disruption facilitates the bidirectional exchange of proteins between the hemolymph and saliva, thereby explaining the presence of proteins not produced in the SGs. These findings are consistent with a previous report that found damage to salivary glands during sporozoite invasion[24]. Intriguingly, the dextran accumulation was dependent on parasite infection, but independent of the parasite loads within the SGs, with some heavily infected SGs displaying no signal or localized dextran invasion (Fig. 5g and Supplementary Fig. 9b).

Next, we performed transmission electron microscopy (TEM) to observe the effect of sporozoite infection on saliva. In TEM images, the saliva appears as a dense, homogeneous matrix inside the secretory cavities (Fig. 5f). Interestingly, in iSGs, the saliva often became granular with electron-lucent spaces, potentially due to the leakage of saliva proteins outside the secretory cavities and/or coagulation of the saliva as it comes in contact with the hemolymph clotting factors (Fig. 5f). This phenomenon was confirmed by two additional laboratories in *A. stephensi* SGs infected with *P. berghei* (Supplementary Fig. 6a–c). Moreover, we detected significant ultrastructural changes in the

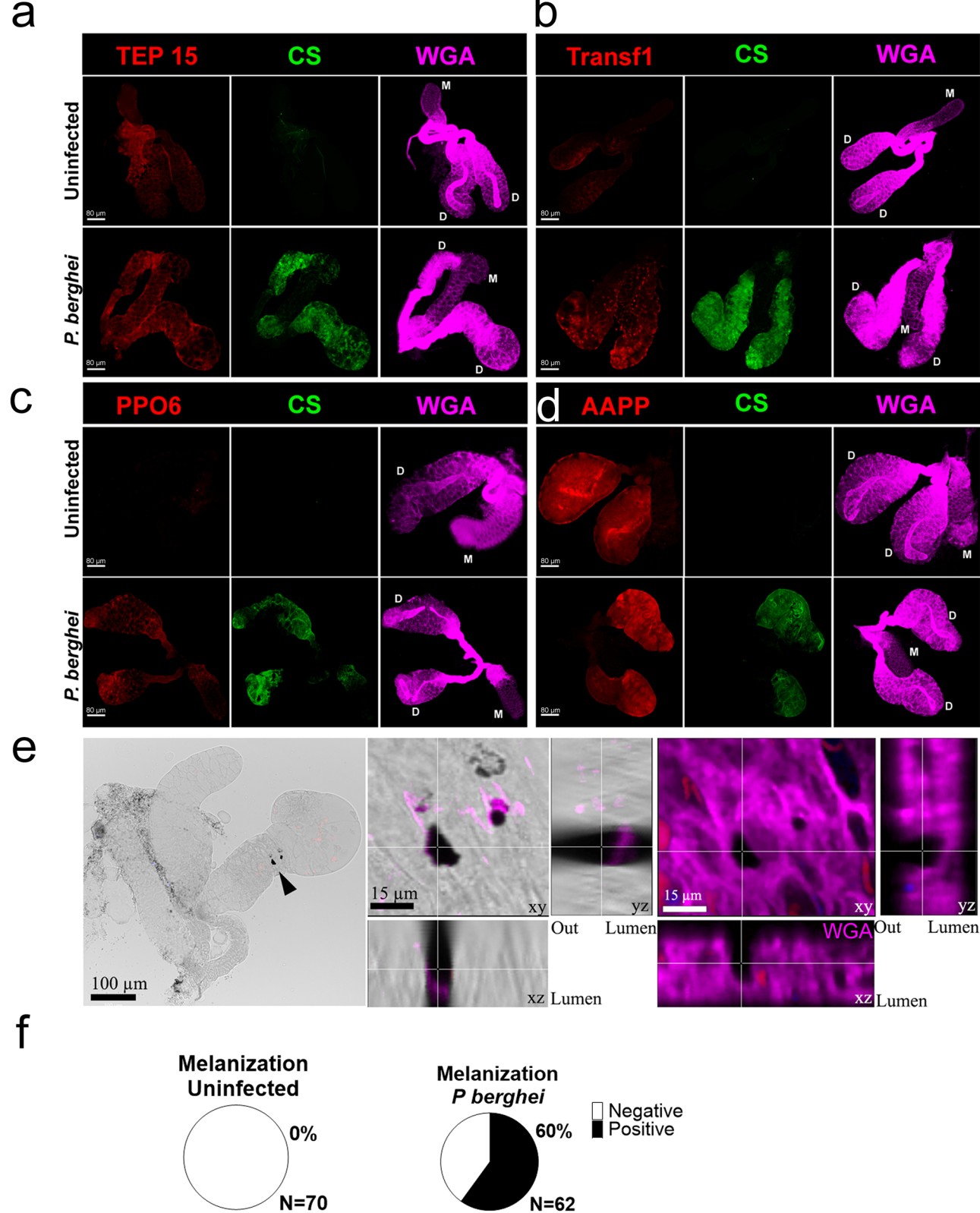

contiguous membranes forming the intercellular contact. While in the uSGs, adjacent acinar-cell (AC) membranes abut tightly, in iSGs the adjacent membranes disengaged, forming a conspicuous intercellular cleft (Fig. 5i); the intercellular space was frequently observed only in iSGs (Supplementary Fig. 6d). Further studies are needed to assess the impact of these morphological changes on saliva secretion, parasite development, and transmission.

We confirmed the changes in the salivary gland morphology and saliva aspect using expansion microscopy. Salivary glands were stained with the protein density marker N-hydroxysuccinimide ester. In iSGs, 94% of the secretory cavities containing sporozoites showed a heterogeneous saliva pattern, while the secretory cavities lacking sporozoites were homogenous (Supplementary Figs. 7a and e). Moreover, 65% of iSGs showed significant structural damage, with loss of salivary

**Fig. 3 | Detection of differentially expressed salivary gland proteins. a–d** SGs were stained for different proteins and visualized using confocal microscopy (*n* = 3 independent experiments). Thioester-containing protein 15 (TEP 15, **a**), transferrin 1 (Transf1, **b**), prophenoloxidase 6 (PPO6, **c**), and *Anopheles* antiplatelet protein (AAPP, **d**). Additionally, *P. berghei* circumsporozoite protein (CS) and wheat germ agglutinin (WGA) were used to visualize infected tissues and SG cell surface, respectively. D: distal lobes, M: medial lobe. Scale bar (**a–d**) 80 μm. **e** Melanization in salivary glands. Bright-field imaging (panel 1) displays the whole salivary gland.

Panel 2 shows sections of the salivary glands, with the white line indicating the focal plane of the images. Panel 3 shows fluorescence images of panel 2, WGA staining in magenta, marking the surface of the salivary gland; sporozoites, stained with mCherry in red; cell nuclei, stained with Hoechst, are shown in blue (*n* = 4 independent experiments). **f** Prevalence of melanization spots in uninfected and iSGs (combined data from *n* = 4 independent experiments with a total of 70 SGs in the control and 62 in the infected group). Source data are provided as a Source Data file.

cavity definition (Supplementary Figs. 7d and 8a). To determine if the observed morphological changes were due to sporozoite invasion, we used sporozoite knockouts for the rhoptry protein RON11 (RON11^cKD)[54]. These parasites develop sporozoites normally, but the majority (>99%) fail to invade the salivary glands. RON11^cKD sporozoites are unable to invade through the typical tight junction formation and therefore attempt to invade the intercellular junctional regions of the gland but fail to enter the salivary cavities (Supplementary Figs. 7b and c). In RON11^cKD iSGs, the SG cavities with heterogeneous salivary patterns were drastically reduced to 21% (Supplementary Figs. 7b, c and e), and the damaged SGs were limited to 5% (Supplementary Fig. 7e and 8a). To objectively measure saliva heterogeneity, we performed Fourier transformations on images of secretory cavities containing or lacking sporozoites, a process that decomposes pixel patterns into frequency components, allowing for the quantification of texture features. Fourier transformations of saliva from infected secretory cavities showed enhanced textural complexity compared to saliva from uninfected secretory cavities (Supplementary Fig. 8b). Collectively, these data indicate that the invasion of sporozoites into the secretory cavity disrupts the structure of salivary glands and disposition of saliva.

## Increased interaction of hemocytes with iSGs

The interaction of hemocytes with SGs in uninfected and infected mosquitoes has not been documented. However, we observed instances of PPO6-positive hemocytes adhering to the surface of SGs (Fig. 4e–g). Interestingly, when performing the dextran diffusion assay, we found hemocytes that had taken up dextran attached to the surface of iSGs (Fig. 5g, and Supplementary Fig. 9). Quantification revealed that iSGs had significantly more attached hemocytes containing dextran than uSGs (Fig. 5h). Importantly, the number of hemocytes attached to iSGs may be underestimated due to limited visual contrast, which complicates the differentiation of hemocytes from adjacent tissues in areas of strong dextran invasion. The increased presence of hemocytes on iSGs likely impacts the detection of proteins on the iSGs proteome. Further studies are required to elucidate the role and molecular profile of these hemocytes.

## Sporozoite-induced changes in the composition of saliva proteins

Mosquito saliva plays a major role in blood feeding and the dynamics of sporozoite transmission[55]. To assess how *Plasmodium* infection and associated salivary gland damage affect saliva composition, we performed label-free quantitative proteomics on saliva samples from uninfected and infected mosquitoes 21 days post-feeding. We performed separate comparative analyses for *P. berghei* and *P. falciparum* infections. Saliva was harvested by inserting the mosquito proboscis into micropipette tips containing 2% pilocarpine, an inducer of salivation, in PBS (Supplementary Fig. 10a). Infected mosquitoes contained between 15,000 and 80,000 sporozoites in their SGs, with a median of 45,000 sporozoites per mosquito (Supplementary Fig. 10b). We identified an average of 85 proteins in the saliva of uninfected *An. gambiae* mosquitoes and 41 in *P. berghei*-infected mosquitoes. Similarly, we identified an average of 106 proteins in uninfected saliva and 69 in *P. falciparum*-infected saliva, with each sample containing at least three unique peptides in at least one group (Fig. 6a, b, and

Supplementary Data 9 and 10). Functional analysis, based on relative protein abundance, revealed that 63 to 73% of proteins in uninfected saliva possessed signal peptides, compared to 58% in *P. berghei*-infected saliva and 75% in *P. falciparum*-infected saliva (Fig. 6c, d). Gene ontology analysis revealed that secretory proteins, which include classical saliva proteins typically found in the hematophagous insect sialomes, were the most prevalent category in all groups (Fig. 6e, Supplementary Data 11).

Principal component analysis revealed two distinct clusters, clearly separating infected from uninfected saliva for both *P. berghei* and *P. falciparum* (Fig. 6f, g). For experiments with *P. berghei*, 26 proteins were exclusively found in the uninfected saliva, whereas 65 proteins were common to both groups. For experiments with *P. falciparum*, 25 proteins were unique to uninfected saliva, and 87 were found in both replicates (Fig. 6h and Supplementary Data 9 and 10). Proteins were classified as differentially enriched or depleted if exclusively detected in one condition or showed statistically significant differences in expression between conditions. Among proteins common to both conditions, 22 proteins were depleted in *P. berghei*-infected saliva (Fig. 7a and Supplementary Data 12), and 43 were depleted in *P. falciparum*-infected saliva (Fig. 7b and Supplementary Data 13). Interestingly, a strong trend of protein depletion was observed, with no proteins enriched in saliva from iSGs. Analysis of the differentially enriched or depleted proteins revealed a notable trend: several hemolymph proteins that were enriched in the SGs following sporozoite invasion exhibited depletion in the saliva collected from infected mosquitoes (Figs. 2d, 7c, d, and Supplementary Data 7, 12, 13). This pattern was particularly evident in immune proteins such as PPO6, TEP15, and CLIPA14.

Among the classical saliva proteins identified, three were depleted in *P. berghei*-infected saliva, while twelve were depleted in *P. falciparum*-infected saliva (Fig. 7e, f, and Supplementary Data 12-13). The depletion of these proteins may facilitate sporozoite transmission by prolonging probing time and increasing the biting rate of infected mosquitoes. Further studies are needed to ascertain the effects of differential expression of these salivary proteins on blood feeding and pathogen transmission. Additionally, even among proteins that were not significantly depleted, a general trend toward decreased protein level was observed (Fig. 7e, f). This combined decrease in protein levels may hinder blood feeding, leading to longer probing times and more bites required for successful feeding, thereby increasing the probability of sporozoite inoculation and transmission.

## Humoral proteins attach to the surface of sporozoites inside the salivary gland

Most sporozoites that invade the SGs remain within this tissue throughout the mosquito lifespan and are never transmitted to a new host. The attachment of specific hemolymph proteins enriched in the whole SG proteome, such as PPO6 and CLIPA14, to the sporozoite surface (Supplementary Fig. 3a, 3d) could partly explain the depletion of these proteins in the saliva proteome: their binding to the sporozoite could reduce their presence in the saliva. To confirm this, we used immunohistochemistry and immuno-transmission electron microscopy. First, we performed an IFA to detect the binding of hemolymph proteins to the surface of sporozoites harvested from the

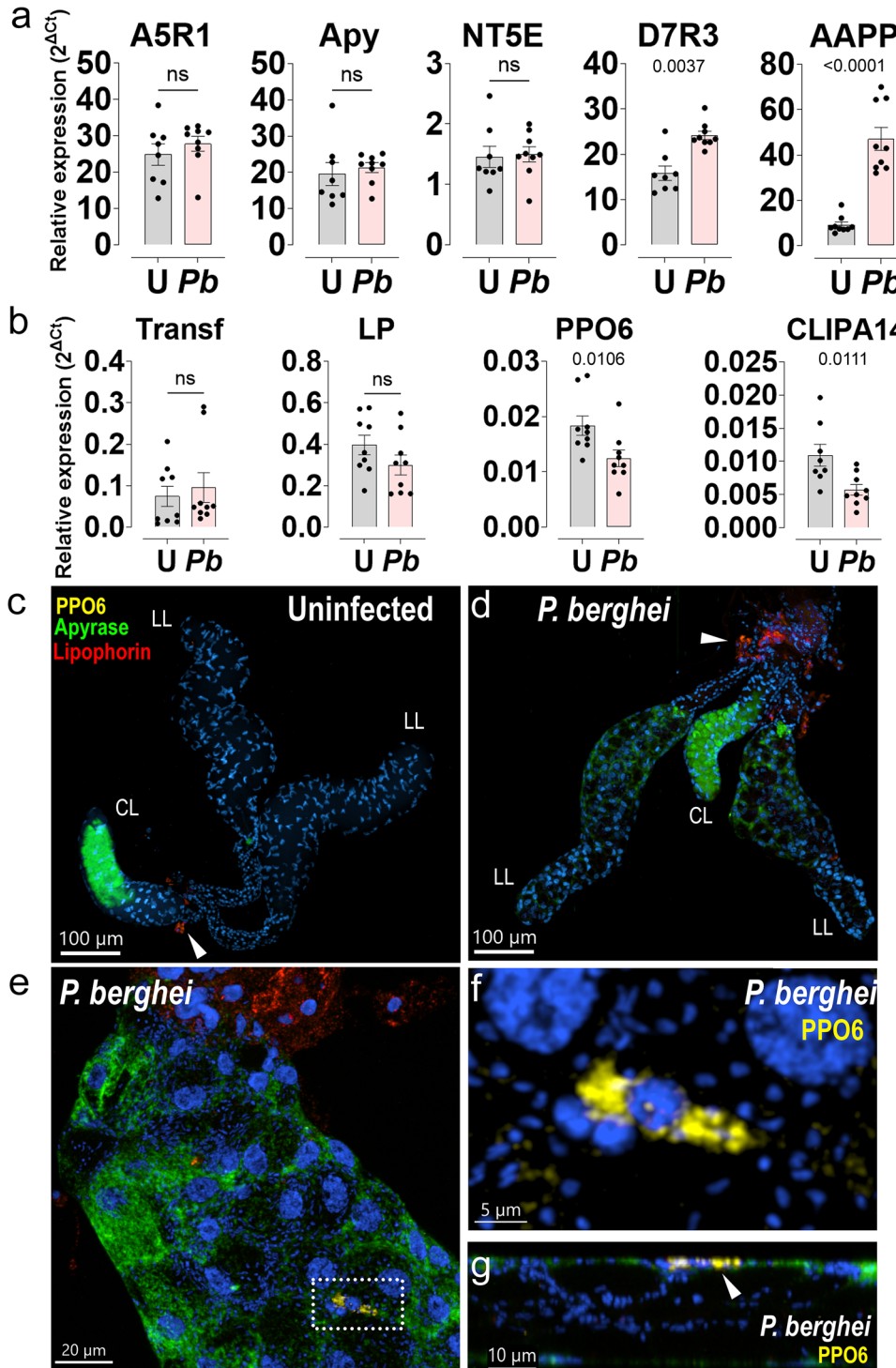

**Fig. 4 | Transcriptional activity in uSGs (U) and *P. berghei*-iSGs (*Pb*). a** Relative mRNA expression of selected classical saliva genes by qRT-PCR analyses in uSGs and iSGs 20 days post-infection. Antigen 5-related protein 1 (A5R1), apyrase (Apy), *Anopheles* antiplatelet protein (AAPP), 5'nucleotidase ecto (NT5E), D7-related protein 3 (D7R3). Relative expression levels were calculated using the delta-delta Ct ($\Delta\Delta Ct$) method, with ribosomal protein S7 used as the reference gene for normalization. **b** Selected proteins showing differential expression between uSGs and iSGs. Transferrin 1 (Transf), lipophorin (LP), clip domain serine protease related protein A 14 (CLIPA14), and prophenoloxidase 6 (PPO6). **a, b** Bars: Mean fold change in gene expression relative to uninfected controls ± SD. Dots: Individual biological replicates. Differences in relative expression between the conditions were determined using an unpaired, two-sided Mann–Whitney U test (Wilcoxon rank-sum). Significance levels are indicated as: $P_{D7R3} = 0.0037$, $P_{AAPP} < 0.0001$, $P_{PPO6} = 0.0106$, $P_{CLIPA14} = 0.0111$. Data derived from three independent experiments with at least two biological replicates. Each dot represents a poll of 15 pairs of salivary glands. **c–g** RNA in situ hybridization in uSGs (**c**) and (**d**) iSGs (*n* = 2 independent experiments). Nuclei in blue, PPO6 in yellow, apyrase in green and lipophorin in red. (LL) lateral lobe and (CL) central lobe. Arrow indicates fat body cells associated with SGs. **e** Close-up of an iSG. Nuclei in blue, PPO6 in yellow, apyrase in green and lipophorin in red. Dashed insert identifies a PPO6-positive hemocyte in the surface of an iSGs. **f** Close-up of a PPO6-positive hemocyte from panel **e**. **g** Lateral view of panel **f**. Arrow indicates hemocyte in the surface of the lobe. Source data are provided as a Source Data file.

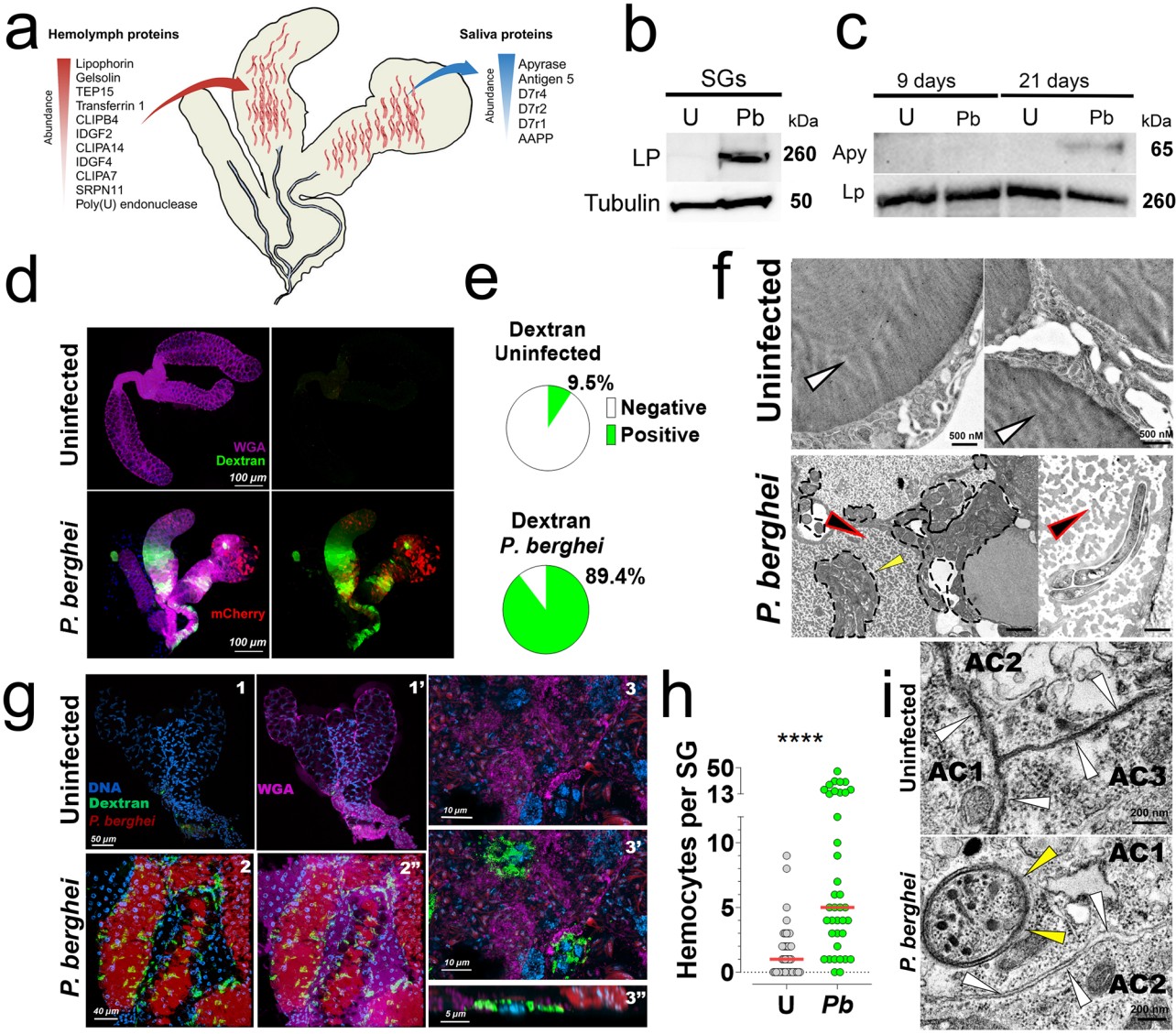

**Fig. 5 | Epithelial integrity in uninfected and _P. berghei_-infected SGs. a** Diffusion of proteins into and out of SGs. Left: hemolymph proteins enriched within iSGs, right: saliva proteins found in infected hemolymph at 19 days post-infection (dpi). Arrows: direction of diffusion. **b** Lipophorin (LP) and tubulin (loading control) western blot on uSGs (U) and iSGs (Pb) at 21 dpi ($n = 2$ independent experiments). **c** Apyrase and Lp (loading control) western blot on uSGs (U) and iSGs (Pb) at 9 and 21 dpi. No SG invasion is detected at 9 days; whereas SGs are heavily infected by 21 days. Lipophorin is used as a loading control ($n = 2$ independent experiments). **d** Dextran diffusion assay in SGs 21 dpi. Dextran (green), mCherry parasites (red), and WGA (magenta) ($n = 2$ independent experiments). **e** Dextran diffusion quantification ($n = 2$). Group differences were tested with a chi-square test ($\chi^2 = 51.14$; $p < 1 \times 10^{-5}$). **f** Representative images of saliva texture within the secretory cavities. Black arrow: electro-lucent areas and granular texture in iSGs, white arrow: homogeneous saliva texture in uninfected SGs, black dashed area: sporozoites

($n = 3$ independent experiments). **g** Fluorescence images showing interactions between hemocytes and SGs. DNA (blue), dextran (green), mCherry (red), and WGA (magenta). uSGs: 1 and 1′, iSGs: 2 to 3″. WGA was omitted in 1 and 2. Merged colors: 1′ and 2″. Closer view of iSGs: 3, 3′. Lateral view of iSGs: 3″ ($n = 2$ independent experiments). **h** Quantification of SG-bound hemocytes: uSGs (U) ($N = 44$) and iSGs (Pb) ($N = 38$). The graph represents an aggregate of two independent experiments. Horizontal red line: median, ****$P \leq 0.0001$ (two-sided Mann-Whitney test). **i** Intercellular-junction morphology. Adjacent acinar cells (AC) in uSGs display continuous, electron-dense junctional complexes (white arrowheads) with opposing membranes fully apposed, and no visible intercellular cleft. In iSGs, junctions appear discontinuous; the plasma membranes disengage, forming a conspicuous intercellular space (white arrowheads). Yellow arrowheads indicate sporozoites ($n = 2$ independent experiments). Source data are provided as a Source Data file.

hemolymph 17 days post-infection. We selected proteins known to circulate at high concentrations in the hemolymph and to be enriched in iSGs. Transferrin 1 bound firmly to the sporozoites, whereas lipophorin, PPO6, and CLIP14A bound faintly; no binding was detected for TEP15 and vitellogenin (Supplementary Fig. 11a). Notably, although lipophorin abounded within the iSGs cavities (Supplementary Fig. 2), its binding to sporozoites was scant. These findings support the idea that hemolymph proteins primarily enter the iSGs through damaged epithelial barriers, rather than by associating with the sporozoite

surface. To test binding within the salivary glands, we selected three hemolymph proteins enriched in the whole SG proteome but depleted in the saliva proteome − TEP15, PPO6, and transferrin, for immunofluorescence analysis. Additionally, we examined AAPP, a saliva protein consistently detected in both uSGs and iSGS proteomes (Fig. 2D). Our analysis revealed that TEP15, PPO6, and transferrin were closely associated with the sporozoite surface (Fig. 8). In contrast, AAPP was dispersed throughout the SGs cavities rather than located on the parasite surface (Supplementary Fig. 12). We also tested whether these proteins

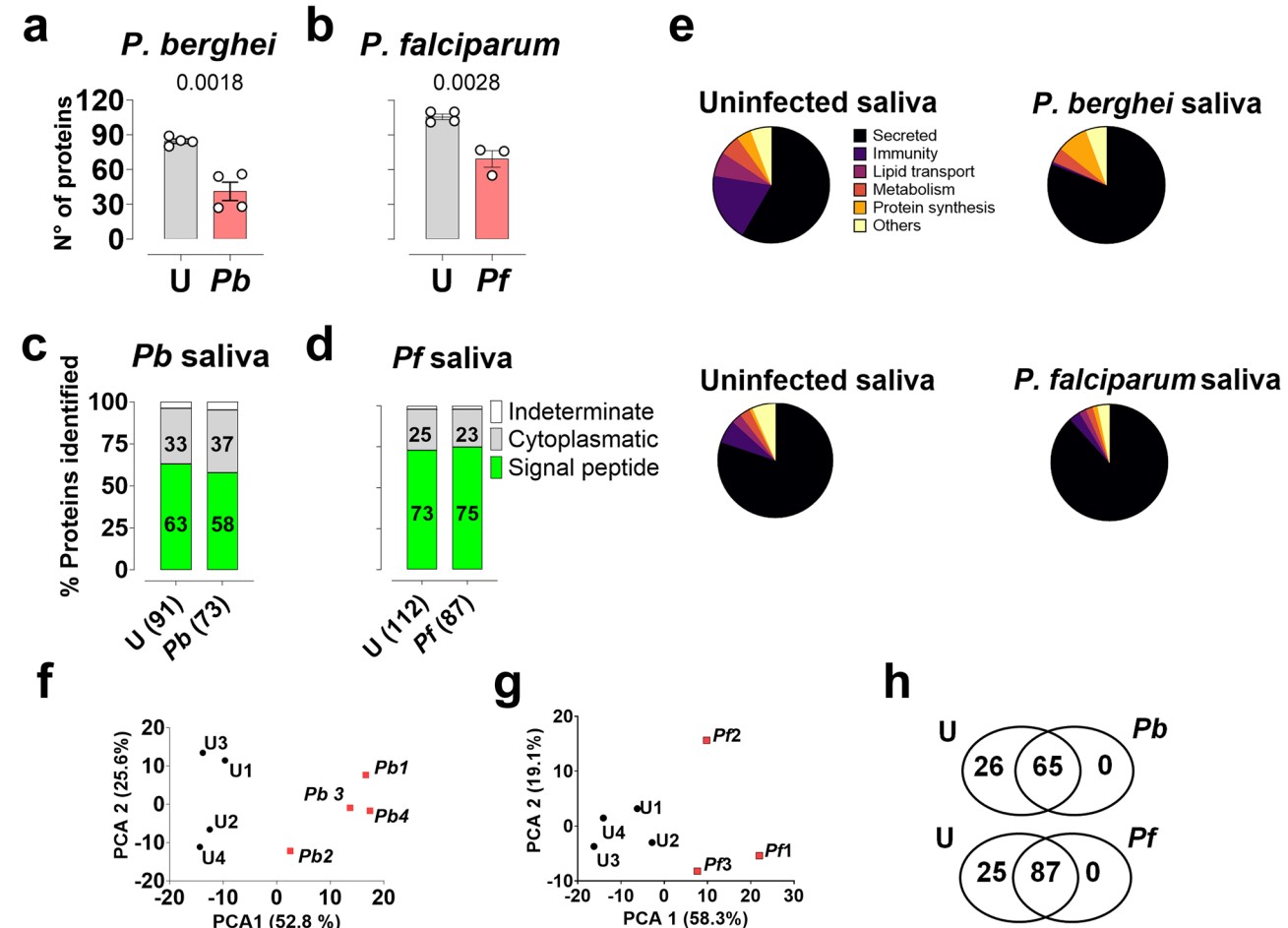

**Fig. 6 | Proteomic analysis of uninfected (U), *P. berghei* (Pb)- and *P. falciparum*-infected saliva (Pf).** Total proteins identified using MaxLFQ in the saliva of *P. berghei* (**a**) and *P. falciparum* (**b**) infected mosquitoes. For the *P. berghei* experiments, analyses were conducted on four independent pools (*n* = 4), each containing saliva from 50 mosquitoes uninfected or infected, whereas *P. falciparum* experiments included four and three independent pools for uninfected and infected tissues, respectively (Uninfected *n* = 4, infected *n* = 3). For **a** and **b**, the data are represented as mean ± SD with a two-sided t-test (P_P. berghei = 0.0018, P_P falciparum = 0.0028). **c**, **d** Classification of proteins predicted by SignalP 5.0, shown as percentage relative iBAQ. **e** Relative protein abundance of functional classes in the saliva proteome of uninfected and infected mosquitoes, expressed as a percentage based on iBAQ values. **f**, **g** Principal component analysis (PCA) of normalized protein abundances based on MaxLFQ values for uninfected and infected saliva. Different colors represent different experimental conditions, and labels identify individual samples. **h** Venn diagram of protein identification. The total number of proteins identified in independent samples (*n* = 4) is plotted as a Venn diagram, showing unique and common proteins between uninfected and infected conditions for each parasite. Source data are provided as a Source Data file.

remained attached to sporozoites after isolation from SGs. Interestingly, only PPO6 exhibited a faint residual binding. Transferrin 1 and TEP15 dissociated from the sporozoite surface after the parasites were removed from the SGs; AAPP, an abundant salivary protein, was not detected attached to the parasite (Supplementary Fig. 11b). These results suggest that these proteins are loosely bound to sporozoites or accumulate around the sporozoites, only while they reside within the SG cavities. This pattern supports the hypothesis that proteins within the cavities invaded by sporozoites are less likely to be inoculated with the saliva.

## Discussion

The sporozoite colonization of mosquito SGs is a critical yet often overlooked phase of the malaria parasite's lifecycle. This stage represents a bottleneck for sporozoite transmission and, thus, offers a promising target for malaria prevention. However, fundamental questions remain regarding how SGs respond to sporozoite invasion and how sporozoites survive within SGs for extended periods. Our research reveals that sporozoite invasion induces morpho-physiological changes in the gland, altering the composition of saliva proteins and potentially influencing malaria transmission (Fig. 9). Notably, both *P. berghei* and *P. falciparum* infections triggered similar changes in the saliva proteome, reinforcing the role of *P. berghei* as a robust model for studying sporozoite-SG interactions. Additionally, while earlier transcriptome studies cataloged the genes expressed in *An. gambiae* SGs, earlier proteomic analyses using 2D gels lagged, identifying only 69 to 122 unique proteins[17,56]. In contrast, using high-resolution proteomics, we significantly expanded this catalog to 3497 mosquito and 277 parasite proteins, providing new insights into *Anopheles*-parasite interactions.

We found that sporozoites that reach the secretory cavities may have access to proteins that transport essential nutrients, including lipophorin and transferrin. In mosquitoes, *Plasmodium* oocysts and sporozoites interact with lipophorin, an important lipid source for the parasite[10,35]. Lipophorin knockdown reduces oocyst numbers and size and produces less infectious sporozoites[10]. Interestingly, *Plasmodium* oocysts seem to modulate lipid metabolism in mosquitoes by releasing an unknown factor that increases lipophorin expression[57]. Another nutrient that sporozoites may access is iron. Transferrin, a glycoprotein that shuttles iron among different tissues, was observed binding

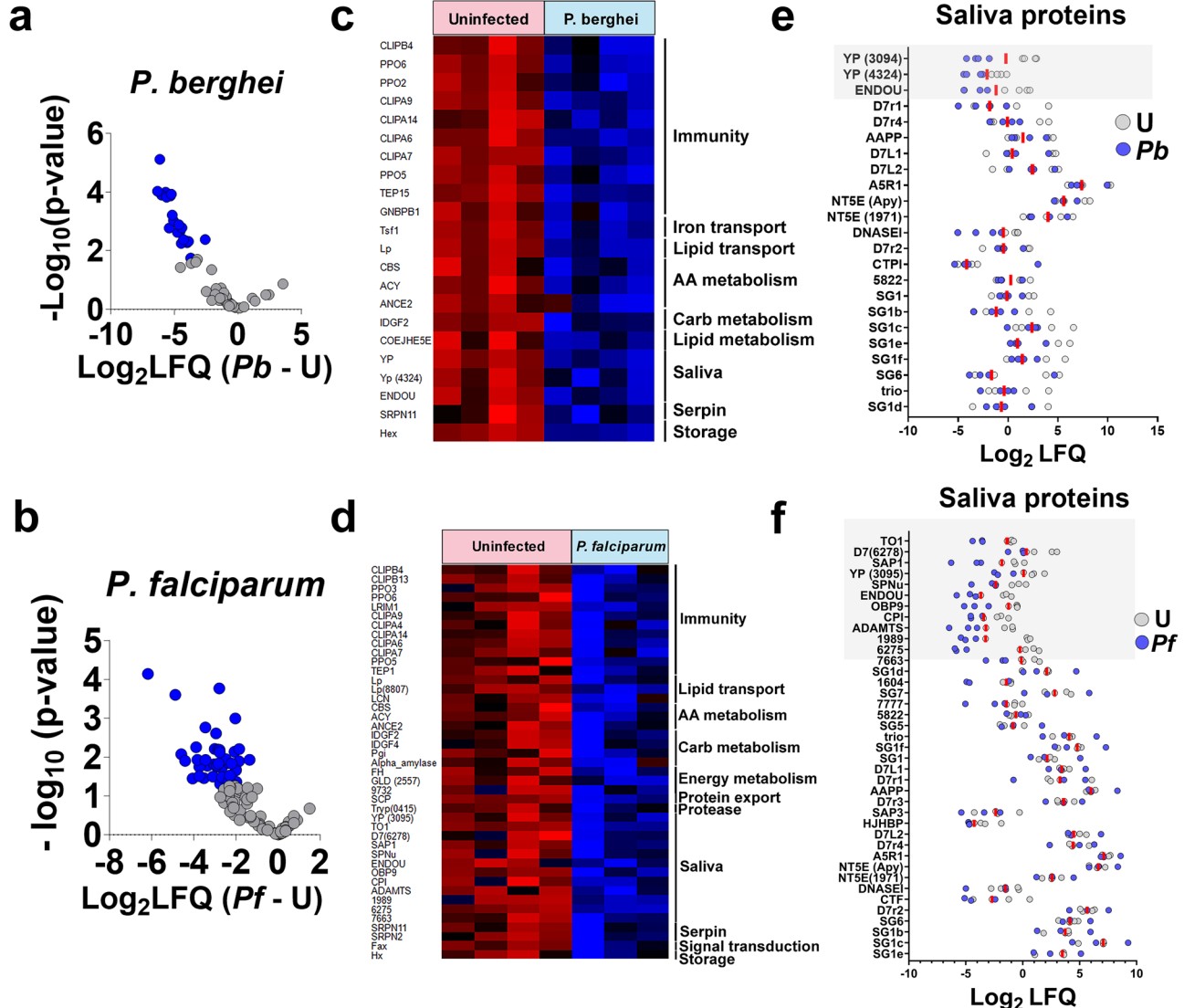

**Fig. 7 | Differential protein enrichment or depletion in uninfected and *P. berghei*- or *P. falciparum*-infected saliva.** Volcano Plot of differential protein expression between uninfected (U) and *P. berghei*-infected saliva (*Pb*) (**a**) and uninfected and *P. falciparum*-infected saliva (**b**). Differential expression was tested with a two-sample Student's t-test with permutation-based FDR control (q ≤ 0.05) using a SAM-style variance parameter ($s_0$ = 0.1). The x-axis represents log2 fold changes, while the y-axis shows significance (−log10 p-value). Proteins depleted in infected saliva are marked in blue, and proteins enriched are marked in red. Only proteins detected in both conditions are included. Heatmap of differentially expressed proteins between uninfected and *P. berghei*-infected (**c**), and uninfected

and *P. falciparum*-infected saliva (**d**). Protein expression values were Z-transformed and normalized across conditions. Rows (proteins) are ordered by function and clustered using Euclidean distance. Dot plot showing the abundance of classical saliva proteins in uninfected and *P. berghei*-infected (**e**), and uninfected and *P. falciparum*-infected mosquitoes (**f**). Each dot represents Log2 LFQ values for individual proteins under uninfected (gray) and infected (blue) conditions. Proteins are listed on the vertical axis, and Log2 LFQ values on the horizontal axis indicate relative abundance. The shadowed areas indicate abundant proteins differentially expressed. Source data are provided as a Source Data file.

to the sporozoite surface within the SGs. Moreover, in other models, transcription of transferrin increases after mosquito infection, suggesting a potential role in the immune response[58]. In *Drosophila*, transferrin participates in nutritional immunity by sequestering iron, thereby hindering pathogen access to this micronutrient[37]. Further studies are needed to determine whether sporozoites exploit host nutrients during the salivary gland stage or if hemolymph proteins enriched in the salivary glands regulate nutrient mobilization. Addressing this question will require tissue-specific downregulation of key metabolic pathways during sporozoite invasion, followed by functional assays to assess parasite viability. Emerging technologies are being developed to enable such investigations.

Previous transcriptome studies reported the upregulation of immune genes in the iSGs, implying that the SGs are

immunocompetent[9]. A limitation, however, was the absence of biological evidence confirming the origin of these mRNAs. This is important, as SGs may be contaminated with other tissues during dissection, such as fat bodies and hemocytes, which respond to infection. The fact that the hemolymph immune proteins upregulated in the proteome of iSGs were not upregulated in the RT-qPCR and in situ hybridization analysis suggests that these proteins most likely entered the SGs due to structural damage to the epithelial layer (Fig. 9), as observed in our dextran diffusion assay and microscopies and as previously reported[24,53]. However, we cannot exclude the possibility that iSGs may import some of these proteins via a specific, yet unidentified, mechanism (pinocytosis or receptor-mediated transcytosis). Moreover, we found that hemocytes are attracted to the iSGs, potentially contributing to the detection of immune proteins in our proteomic

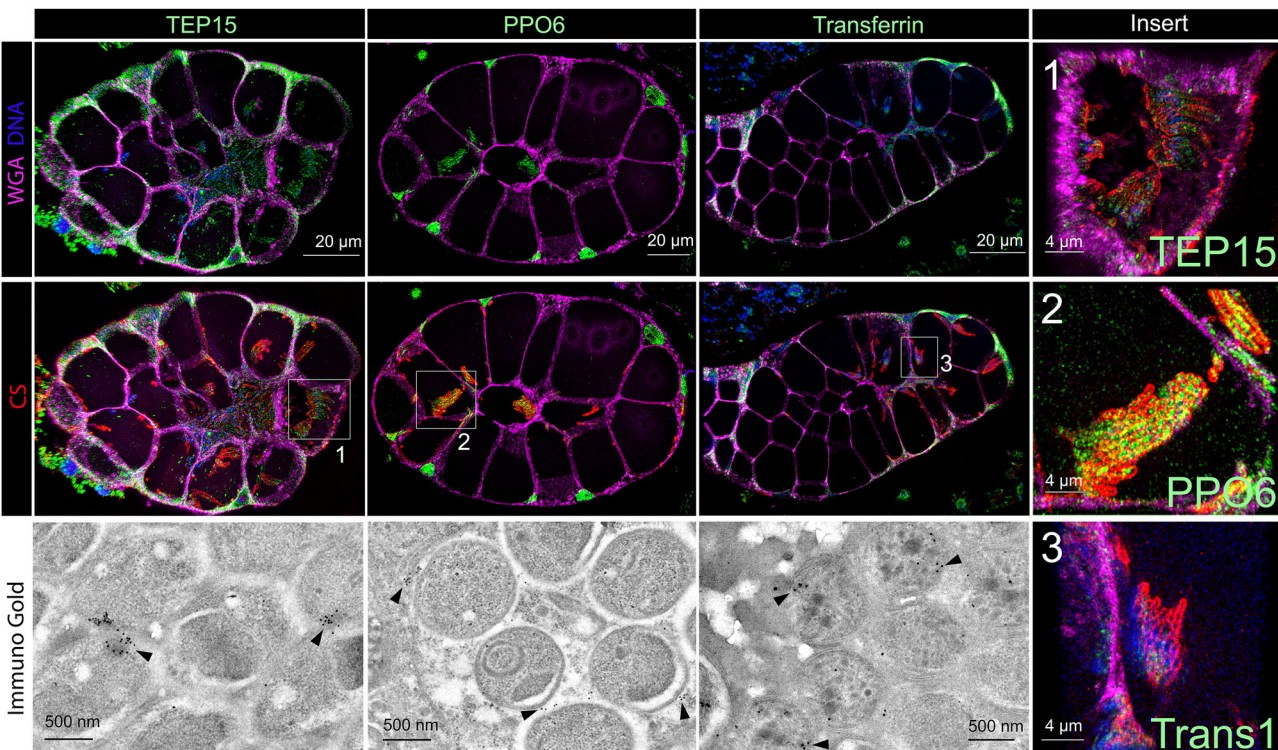

**Fig. 8 | Histological immunofluorescence and immuno-transmission electron microscopy of *P. berghei*-iSGs.** These images display the distribution and association of immune proteins with the surface of *P. berghei* parasites in the SGs. Staining includes TEP15, PPO6, transferrin (green), circumsporozoite (CS, red), and WGA (magenta). Insert provides a closer view of stained *P. berghei* parasites, highlighting the spatial relationship between immune proteins and the parasite surface. Bottom images: immuno-TEM images showing the detailed interaction between immune proteins and *P. berghei* parasites − arrowheads showing immunogold staining of TEP15, PPO6, and transferrin, emphasizing the close association of immune proteins with the parasite surface (*n* = 3 independent experiments for immunofluorescence, *n* = 1 transmission electron microscopy).

analysis. Though novel, this finding is unsurprising, as mosquito hemocytes patrol epithelial tissues searching for pathogens[59]. Furthermore, these immune cells may contribute to tissue regeneration, as observed in *Drosophila* and mammals, specifically M2 macrophages[60,61]. We observe that some immune-related hemolymph proteins enriched in infected salivary glands, including TEP15, PPO6, and transferrin, are attached to the sporozoite surface. This suggests that certain hemolymph proteins may enter the salivary glands while bound to sporozoites. However, lipophorin, a high-molecular-weight protein (366 kDa), does not associate with hemolymph sporozoites. Given its size and lack of sporozoite binding, it is highly likely that lipophorin and other salivary proteins enter the glands via diffusion, driven by structural damage induced by sporozoite invasion.

Our findings suggest that sporozoites within the salivary gland are closely associated with hemolymph proteins, especially those from the phenol-oxidase pathway. This association raises the question of how the sporozoites evade melanization within the SGs. A known evasion mechanism for hemolymph sporozoites involves post-translational modifications of cell surface proteins by the parasite's glutaminyl cyclase[43]. However, this mechanism does not account for sporozoites in the SGs, as knockout parasites within the SGs were not melanized, supporting the idea that the SGs are an immunoprivileged tissue[43]. Another possibility is that the parasite recruits factors that inhibit melanization. In iSGs, we detected an enrichment of CLIPA14, which was bound to the parasite surface. CLIPA14 acts as an agonist for *Plasmodium*, inhibiting the PPO pathway[62,63]. Thus, the parasite may recruit negative regulators like CLIPA14 and CLIPA7 to prevent PPO activation by agonists like CLIPB4.

Saliva proteins are crucial to vectorial capacity, influencing blood feeding, oocyst development, and sporozoite transmission[55,64,65].

Therefore, we investigated whether changes in the salivary gland proteome affect saliva composition. Surprisingly, we found that the humoral hemolymph proteins enriched in iSGs were depleted in the saliva of *P. berghei*- and *P. falciparum*-infected mosquitoes, with many being immune proteins adhering to the parasite inside the glands. Notably, for both parasites, we observed a reduction in yellow proteins and EndoU and, for *P. falciparum*, a decrease in one D7 protein. All other saliva proteins tended toward depletion, including apyrase. The overall reduction in salivary proteins may impact parasite transmission dynamics by increasing probing time and bite frequency, similar to the effects observed when apyrase and D7 levels are reduced, ultimately facilitating sporozoite transmission[16,22].

We observed that salivary gland invasion by sporozoites causes epithelial damage, resulting in the leakage of saliva proteins into the mosquito hemolymph. This leakage may affect mosquito physiology, fitness, and fertility, as it includes enzymes, protein inhibitors, and kratagonists−proteins that sequester signaling molecules−potentially disrupting key signaling pathways. Further investigation is needed in this area. Previous studies show that sporozoites cause minimal damage to invaded SG cells by utilizing the AMA1-RON complex to form a vacuole with a ring-like structure that mimics a moving junction[53]. This is essential to maintain the general structure of the SG by preventing the rapid loss of salivary gland cells and the collapse of the SG. However, our results show that although sporozoites do not induce the massive loss of SG cells, the invasion disrupts the structural integrity of the tissue, leading to hemolymph uptake, and saliva leakage as previously suggested[24] (Fig. 9). Mosquito hemolymph contains clotting factors[66], making it plausible that hemolymph entering the salivary gland during sporozoite invasion causes saliva to clot in specific areas, thereby reducing the gland's functional capacity. This could

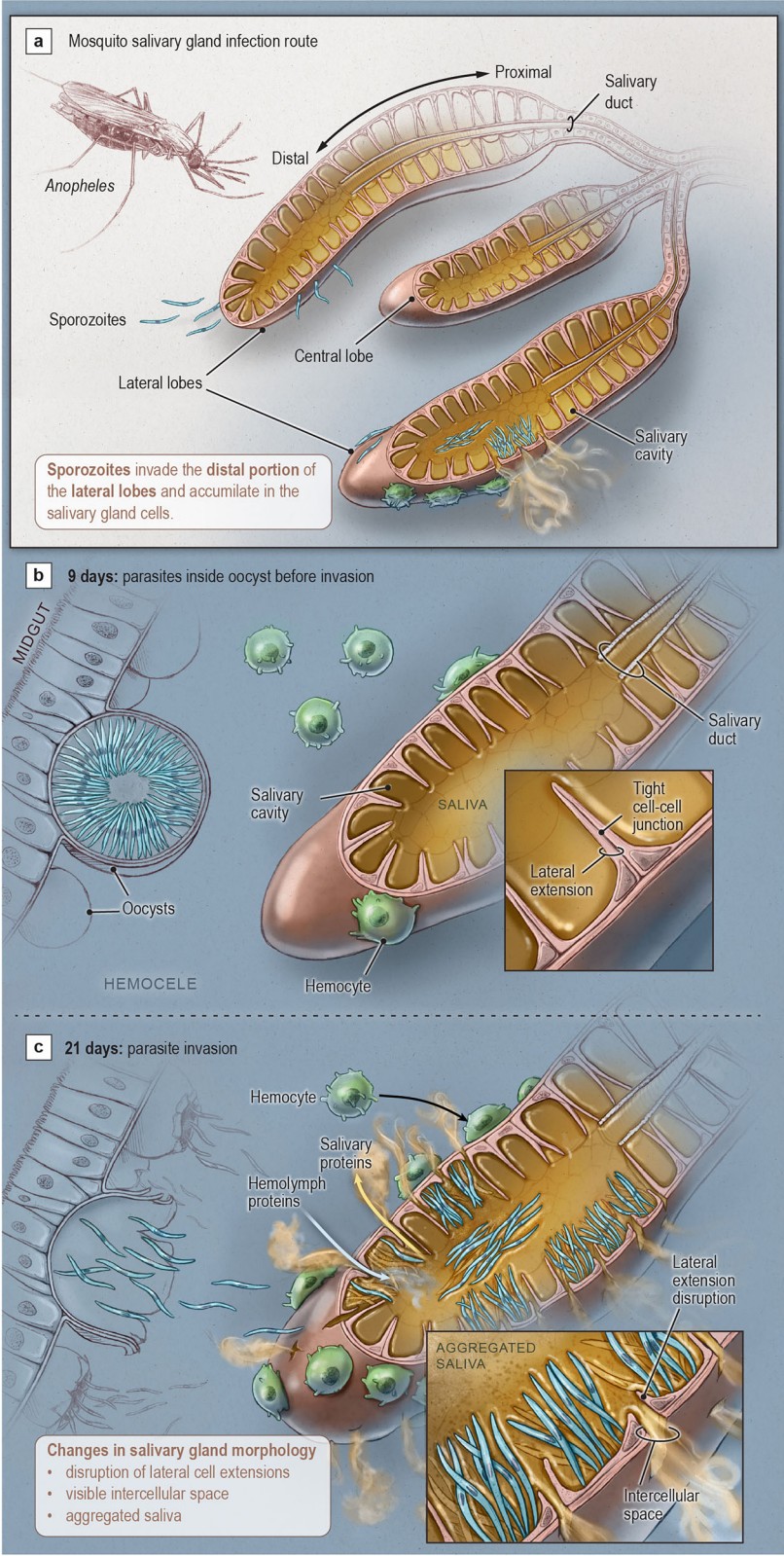

explain the altered saliva texture observed in cavities invaded by sporozoites. Some studies have noted that infected mosquitoes exhibit increased probing time and decreased apyrase activity, previously described as a "biochemical lesion"[16,20]. If hemolymph is indeed clotting the saliva, it may prevent the release of proteins in those regions, leading mosquitoes to probe more and deliver more sporozoites to the skin during biting. While these hypotheses offer potential explanations

for our observations and previous findings, further research is needed to fully understand the implications of saliva proteins entering the mosquito hemolymph, particularly regarding parasite transmission.

Our study provides an updated catalog of protein expression in *An. gambiae* SGs, revealing that transcriptional activity, epithelial damage, and hemocyte attraction influence changes in protein abundance in this tissue. These factors may affect mosquito feeding

**Fig. 9 | Salivary gland invasion by *Plasmodium* sporozoites disrupts gland morphology and alters saliva composition. a** Schematic of mosquito salivary gland (SG) anatomy and sporozoite invasion route. *Plasmodium* sporozoites invade the distal lateral lobes of *Anopheles* SGs, entering the secretory cavities where they accumulate prior to transmission. **b** At 9 days post-infection (dpi), sporozoites remain inside midgut oocysts and have not yet invaded the SGs. SG epithelial cells exhibit tight lateral cell-cell junctions, maintaining tissue integrity. **c** At 21 dpi, sporozoites actively invade the SGs, causing morphological disruptions, including

loss of lateral extensions, formation of visible intercellular spaces, and aggregation of saliva within the cavities. Hemolymph proteins and hemocytes are observed in close association with invaded glands, suggesting compromised epithelial barriers and leakage of saliva and hemolymph components. Insets show intact cell junctions in uninfected glands (**b**) and disrupted cell extensions and cell junctions with aggregated saliva in infected glands (**c**). These structural changes likely impair SG function and alter saliva composition, potentially impacting mosquito physiology and parasite transmission.

behavior and sporozoite transmission. We offer a functional description of sporozoite-salivary gland interactions, setting the stage for future studies on epithelial damage, parasite nutrition, saliva proteins impact on the hemocoel, and hemocyte roles in infection. Using AI or machine learning to analyze involved genes and proteins could identify key pathways for functional studies and interventions, potentially predicting similar interactions in other *Plasmodium*/vector combinations. These insights could advance understanding of parasite-vector interactions and target the salivary gland bottleneck, aiming to prevent sporozoite transmission.

## Methods

### Parasites, mosquitoes, and infections
*Anopheles gambiae* Keele strain[67] was reared at 28 °C and 80% relative humidity, with a 12 h/12 h light/dark cycle under standard laboratory conditions. Adult mosquitoes were fed *ad libitum* at 10% sucrose solution.

### Plasmodium berghei infection
For the proteome, we performed infections with a transgenic GFP *P. berghei* parasite strain (ANKA GFPcon 259cl2). For all the other assays, we used *P. berghei* (Pb) ANKA transgenic line expressing mCherry and luciferase markers under the control of the hsp70 and eef1a promotors, respectively[68]. Parasites were maintained by serial passages in female BALB/c mice. Parasitemia was quantitatively assessed by light microscopy using methanol-fixed, 10% Giemsa-stained blood smears. For mosquito infections, female mosquitoes (4-5 days old) were fed on infected mice once flagellation reached 3-4 exflagellations per 40 x field. After feeding, infected mosquitoes were kept at 19 °C, 80% humidity, and a 12 h light-dark cycle. Mosquitoes with a productive infection in the salivary glands were sorted using a fluorescent scope (Leica M205 FCA coupled to a camera DFC 7000 G5 for multi-color fluorescence imaging). As controls, mosquitoes from the same cohort were blood-fed on uninfected mice at the same time as the infected group.

### Plasmodium falciparum infection
Female *An. gambiae* mosquitoes were infected with *P. falciparum* NF54 via membrane feeding using reconstituted human blood[69]. The asexual blood-stage cultures were maintained in vitro using O+ erythrocytes at a 4% hematocrit, as previously described[69]. For mosquito infections, a suspension of human RBCs was mixed with human serum (Interstate blood bank) at 50% hematocrit and diluted to 0.05% gametocytemia before being fed to the mosquitoes. Post-infection, the mosquitoes were kept at 25 °C and 80% humidity and provided with a 10% (w/v) sucrose solution.

### Salivary gland dissection and saliva collection
For proteomic analysis of mosquito salivary glands, we used mosquitoes infected with *P. berghei* parasite as described above. Three independent infections were conducted for the proteome analysis, each one consisting of three pools of 25 pairs of uninfected or infected salivary glands. Salivary glands were dissected 21 days post-infection and immediately added to RIPA buffer (Thermo Fisher) containing Halt™ Protease and Phosphatase Inhibitor Cocktail (Thermo Fisher).

After dissection, the SGs were immediately snap-frozen in liquid nitrogen and stored at −80 °C until further use.

For saliva collection, forced salivation was used on 50 female mosquitoes per group. Three independent infections with *P. berghei* or four with *P. falciparum* were performed for this assay; either of these parasites was paired with an uninfected sample. The mosquitoes were water-deprived for approximately 1–2 h before collection. After sedation on ice, the mosquitoes' mouthparts were inserted into 10 μl tips containing 8 μl of PBS supplemented with 2% pilocarpine (Sigma-Aldrich). The mosquitoes were then allowed to salivate for 30 minutes at 28 °C. The saliva solutions were subsequently pooled into Protein Lo-Bind tubes (Eppendorf) and stored at −80 °C until further use.

### Hemolymph collection
In two independent experiments, Hemolymph was collected by perfusion from 25 uninfected or *P. berghei*-infected mosquitoes. Mosquitoes were anesthetized on ice and placed individually under a stereoscope on parafilm. An incision was made on the lateral region of the VI-VII abdominal segments using a U-100 insulin syringe. A heat-stretched capillary needle filled with sterile PBS was inserted laterally into the thorax, and approximately 10 μL of sterile PBS was injected to displace the hemolymph. The hemolymph flowed through the abdominal incision and was collected with a pipette (P20, Gilson) into pre-chilled Protein LoBind tubes (Eppendorf) on ice. After collection, a protease inhibitor (Halt™ Protease and Phosphatase Inhibitor Cocktail, Thermo Fisher) was added to a final concentration of 1×.

For proteomic analysis, hemolymph samples were centrifuged at 1000 *g* for 10 minutes at 4 °C. The supernatant (plasma) was transferred to a new Protein LoBind tube, and the pellet was discarded. The plasma was further centrifuged at 10,000 g for 10 minutes at 4 °C to remove prohemocytes and residual cellular debris. The clarified plasma was stored at −80 °C for mass spectrometry.

### Mass spectrometry
The samples dissolved in SDS-PAGE sample buffer were run on an SDS-PAGE minigel. The gels were stained with Coomassie blue (ProtoBlue Safe, National Diagnostics) overnight. The samples for the whole salivary gland proteome and hemolymph proteome were cut into six slices, whereas for the saliva samples, the top gel slices were excised and cut into two pieces. Next, the excised slices were cut into small pieces and subjected to in-gel trypsin digestion. Briefly, the gel pieces were destained with 50% acetonitrile in 100 mM AMBI, reduced with 5 mM DTT, and alkylated in 11 mM iodoacetamide. The pieces were then dehydrated with 50%, followed by 100%, acetonitrile. Afterward, 10 ng/μL mass-spec-grade trypsin (Promega) was added, allowing the trypsin solution to soak into the gel. Following overnight digestion at 30 °C, the peptides were extracted with acetonitrile in formic acid and subjected to LC-MS/MS analysis.

LC-MS/MS data were acquired using an Orbitrap Fusion Lumos mass spectrometer equipped with an EASY-Spray Ion Source and an EASY-nLC 1200 liquid chromatography system (Thermo Fisher Scientific). The mobile phase consisted of water with 0.1% formic acid. Peptides (5 μL) were loaded onto a trap column (PepMap 100 C18, 3 μm particle size, 2 cm length, 75 μm inner diameter, Thermo Fisher Scientific), and separated on an analytical column (PepMap 100 C18,

2 µm particle size, 25 cm length, 75 µm inner diameter, Thermo Fisher Scientific) using a linear gradient: 0–40% acetonitrile over 80–100 min, 40–80% for 5 min, holding at 80% for 5 min, 80–0% for 5 min, and holding at 0% for 5 min. Throughout this 120 min data acquisition, the flow rate was set at 300 nL/min, with the analytical column maintained at 50 °C.

Data acquisition followed a standard data-dependent acquisition strategy. MS1 scans were performed every 2 s using the Orbitrap mass analyzer at a resolution of 120,000. MS2 scans were performed on multiply charged precursor ions, isolated with a 1.6 m/z window using a quadrupole, and fragmented by CID at 35% collision energy, analyzed with the Linear Ion Trap. A dynamic exclusion period of 30 seconds was applied, and EASY-IC internal calibration was used for Orbitrap scans.

### Data processing

LC-MS/MS data were processed using MaxQuant software (v2.0.3.0.). Raw data files from each sample were loaded as fractions and searched against the *Anopheles gambiae* PEST annotated protein database (VectorBase, release 51 for saliva proteome; release 54 for the whole salivary gland proteome, release 68 for the hemolymph proteome) and *Plasmodium berghei* ANKA protein database (PlasmoDB, release 51 for saliva proteome; release 54 for the whole salivary gland proteome, release 68 for the hemolymph proteome), with the MaxQuant contaminants database included. For *Plasmodium falciparum* samples, the *P. falciparum* NF54 protein database (release 51) was used. Peptide identification allowed for fixed cysteine carbamidomethylation and variable modifications of N-terminal acetylation and methionine oxidation. A 1% false discovery rate (FDR) was applied, and the *match between runs* feature was enabled. Protein quantification for differential expression was performed using label-free quantification (LFQ) intensity, which was reported as log2 values in tables and figures. Only proteins with at least three unique peptides were included, and contaminants were excluded from further analysis.

Statistical analyses were performed using Perseus software (1.6.13)[70]. Protein intensities were log2-transformed and normalized by median centering. Only proteins with LFQ values greater than zero were included in the analysis. Missing values were imputed using a random normal distribution, with the mean set to the observed distribution mean minus 1.8 standard deviations and the standard deviation set to 0.3 times the observed standard deviation. Differential expression was assessed using Student's t-test with permutation-based FDR control. Proteins with $q$ value ≤ 0.05 were considered differentially expressed. Heatmaps were generated on Perseus, and PCA plots were generated using GraphPad Prism 9.

### Protein annotation

For protein annotation, we employed an in-house program that analyzes BLASTp and rpsBLAST results across various databases, including UniProtKB, Conserved Domain Database (CDD), REFSEQ-Invertebrate, Diptera, and FlyBase. This program evaluates approximately 400 keywords and their order of appearance in the protein matches, considering e-values and sequence coverage. Signal peptides were predicted using SignalP 5.0. Protein abundance was quantified using intensity-based absolute quantification (iBAQ). Relative abundances within each pool were calculated by dividing individual iBAQ values by the total sum of iBAQ values for that pool[25,26,71].

### Immunofluorescence assays from whole tissue mounting

SGs were dissected from adult female mosquitoes, either uninfected or infected with *P. berghei*, 21 days post-infection. To facilitate handling, SGs were initially kept attached to the mosquito heads and detached just before mounting. The SGs were fixed in 4% paraformaldehyde at room temperature for 1 h. Following fixation, the SGs were washed three times with PBST (PBS containing 0.1% Triton X-100) to remove residual fixative. Blocking was performed by incubating the SGs in PBST supplemented with 5% BSA (bovine serum albumin) for 1 h at room temperature. Primary antibodies were diluted in the blocking buffer as follows: anti-TEP15 rabbit monospecific IgG (1 µg/ml, Pacific Immunology), anti-Transferrin 1 rabbit monospecific IgG (1 µg/ml, Pacific Immunology), anti-PPO6 rabbit polyclonal serum (1 µg/ml) − kindly provided by Dr. Ryan Smith from Iowa State University −, anti-AAPP rabbit monospecific IgG (1 µg/ml, Pacific Immunology), and anti-circumsporozoite (CS) protein mouse monoclonal antibody (3D11, 1:1000). SGs were incubated with these primary antibodies overnight at 4 °C. After primary antibody incubation, SGs were washed three times with the blocking buffer for 20 min each. They were then incubated with secondary antibodies, diluted in the blocking buffer, for 4 h at room temperature. The secondary antibodies used were Alexa Fluor 488-conjugated, Alexa Fluor 594-conjugated goat anti-mouse or goat anti-rabbit antibodies (1:1000, Thermo Fisher). Wheat Germ Agglutinin (WGA) conjugated to Alexa Fluor 647 (Thermo Fisher) at 2 µg/ml was employed to visualize N-acetyl glucosamine residues. Following, the SGs were washed three times in PBST. They were counterstained with Hoechst 33342 (20 µM, Thermo Fisher) for 20 min. The tissues were mounted using ProLong Gold Antifade (Thermo Fisher). Confocal images were obtained using a Leica TCS SP8 DM8000 confocal microscope (Leica Microsystems, Wetzlar, Germany) with a 40× or 63× oil immersion objective. The microscope was equipped with a photomultiplier tube/hybrid detector. Microscope slides were mounted using a drop of Prolong Gold Antifade Mountant (ThermoFisher). Image processing was performed using Imaris 9.9.1 (Bitplane, Concord, MA, USA) and Adobe Photoshop CC (Adobe Systems, San Jose, CA, USA).

### Histopathology and Immunofluorescence

Specimens were fixed in 10% neutral buffered formalin for 1 h then transferred to 70% ethanol prior to suspension in Tissue Guard Gel (TG12, StatLab) according to manufacturer's recommendations. The gel pellet was trimmed then processed in paraffin with an expedited tissue processing protocol on the Leica ASP6025. Samples were embedded into paraffin blocks then sectioned at 4-5 µm.

Staining was achieved on the Bond RX automated system with the Bond Research Detection Kit (DS9455, Leica). Tissue sections were deparaffinized with the Bond Dewaxing Solution (AR9222, Leica) at 72 °C for 30 min then subsequently rehydrated with graded alcohol washes and 1x Bond Wash Solution (Leica). Heat-induced epitope retrieval (HIER) was performed using Epitope Retrieval Solution 1 (Leica), heated to 100 °C for 20 min. Tissues were blocked with Protein Blocker (X0909, Dako) for 30 min prior to a 60 min incubation with the following primary antibodies 1 µg/ml: anti-TEP15 rabbit monospecific IgG(1 µg/ml, Pacific Immunology), anti-Transferrin 1 rabbit monospecific IgG (1 µg/ml, Pacific Immunology), anti-PPO6 rabbit polyclonal serum (1:500) − kindly provided by Dr. Ryan Smith from Iowa State University (Kwon and Smith, 2019, PMID: 31235594) −, anti-AAPP rabbit monospecific IgG (1 µg/ml, Pacific Immunology), anti-Lipophorin monospecific IgG (1 µg/ml, Pacific Immunology), anti-CLIPA14 (1:400, Boster Bio), and anti-circumsporozoite (CS) protein mouse monoclonal antibody (3D11, monospecific IgG). Samples were rinsed with wash solution then incubated for 30 min with the following secondary antibodies: Alexa Fluor 488-conjugated, Alexa Fluor 594-conjugated goat anti-mouse or goat anti-rabbit antibodies (1:1000, Thermo Fisher), Wheat Germ Agglutinin (WGA) conjugated to Alexa Fluor 647 (Thermo Fisher) at 2 µg/ml. Slides were counterstained with Hoechst 33342 (20 µM, Thermo Fisher) for 20 min. All antibodies were diluted in Background Reducing Antibody Diluent (S3022, Agilent). Slides were mounted with ProLong Gold Antifade Mountant (P36934, Invitrogen). Images were processed using Imaris 9.9.1 (Bitplane, Concord, MA, USA) and Adobe Photoshop CC (Adobe Systems, San Jose, CA, USA).

## Transmission electron microscopy and immune staining

Protocol at NIAID: Salivary glands were fixed in 2% paraformaldehyde in 0.1 M cacodylate buffer and stored at 4 °C until further processing. Samples were spun into a pellet for microwave assisted (BioWave, Ted Pella) electron microscopy preparation. Briefly, pellets were rinsed with buffer, dehydrated using a graduated ethanol series, and infiltrated with LR White (Electron Microscopy Sciences). Blocks were polymerized at 50 °C in a vacuum oven overnight. For immune gold labeling: 70 nm sections were placed onto carbon coated 200 hex mesh gold grids. Sections were blocked for 1.5 h at room temperature in 5% BSA, 10% NGS, 1% fish gelatin, and 0.01% Tween-20 in PBS. Sections were placed in primary antibody (1:50 for PP06, TEP15, and TR1) O/N at 4 °C along with a no-primary control (block only). Sections were rinsed with blocking buffer and placed in 1:20 25 nm-goat-anti-rabbit (BBI Solutions, Ted Pella) secondary antibodies for 2 h at room temperature. Sections were rinsed with PBS, followed by dH20, and poststained with 2% aqueous uranyl acetate. Sections were imaged with a Tecnai T12 transmission electron microscope (Thermo Fisher) operating at 120 eV with a Rio digital camera (Gatan).

Protocol at Heidelberg University: salivary glands from uninfected or *P. berghei*-infected *An. stephensi* mosquitoes were dissected on day 19 post infection and immediately fixed using 4% paraformaldehyde and 1% glutaraldehyde in 0.1 M PHEM buffer (60 mM PIPES, 10 mM EGTA, 25 mM HEPES, 2 mM Magnesium sulfate, pH 6.9) over night at 4 °C. Next, using a BioWave Pro+ microwave the glands were further fixed with 1% osmium tetroxide in 0.1 M PHEM buffer and post-fixed in 1% uranyl acetate in ddH2O. The fixed glands were dehydrated in an acetone series and embedded in Spurr's resin set to polymerize at 60 °C for 48 h. Once polymerized, the block was sectioned on a Leica EM UC7 using a DiATOME diamond knife to generate 70 nm sections, which were picked up on formvar coated grids. Salivary gland sections were imaged on a 70 kV Jeol JEM1400 TEM equipped with a TVIPS TemCam F416 4k x 4k pixel digital camera.

Protocol at Johns Hopkins University: For thin-section transmission electron microscopy (TEM), salivary glands of *An. stephensi* mosquitoes containing P. *berghei* sporozoites (ANKA-strain 2.34) were fixed in 2.5% glutaraldehyde (Electron Microscopy Sciences; EMS) in 0.1 M sodium cacodylate buffer (pH 7.4) for 1 h at room temperature. They were washed 3 times in 0.1 M cacodylate buffer and then postfixed for 1 h in 1% osmium tetroxide (EMS) in the same buffer at room temperature. After 3 washes in water the samples were stained for 1 h at room temperature in 2% uranyl acetate (EMS), then washed again in water and dehydrated in a graded series of ethanol. The samples were then embedded in Embed-812 epoxy resin (EMS). Ultrathin (50–60 nm) sections were cut using a Reichert Ultracut ultramicrotome and collected on formvar- and carbon-coated nickel grids, stained with 2% uranyl acetate and lead citrate before examination with a Philips 410 Electron Microscope (Eindhoven, the Netherlands) under 80 kV.

## Dextran diffusion and detection of phenoloxidase activity

Uninfected or *P. berghei*-infected mosquitoes, 9 or 21 days post-infection, were cold-anesthetized and injected with 50 nl of 10 kDa anionic fixable dextran conjugated with Alexa Fluor 488 (1 µg/µl, Thermo Fisher). After injection, mosquitoes were maintained at 26 °C for 3 h to allow the dextran to circulate in the hemolymph. SGs were dissected from the injected mosquitoes and fixed in 4% paraformaldehyde for 1 hour at room temperature in nine-well excavated glasses. The SGs were then washed twice with PBS to remove residual fixative and counterstained with Wheat Germ Agglutinin (WGA) 647 (2 µg/ml, Thermo Fisher) and Hoechst 33342 (20 µM, Thermo Fisher) for 30 min. Finally, the samples were mounted with ProLong Gold Antifade Mountant (Thermo Fisher) for imaging. To detect phenoloxidase activity, the prevalence of melanization spots was computed from two independent experiments.

## Hemocyte staining

Uninfected mosquitoes were cold-anesthetized and injected with 50 nl of 10 kDa anionic fixable dextran conjugated with Alexa Fluor 488 (1 µg/µl, Thermo Fisher). After injection, mosquitoes were maintained at 26 °C for 3 h. The hemolymph of 15 mosquitoes (2 µl per mosquito) was collected by perfusing the mosquito abdomen with anticoagulant buffer (60% Schneider medium, 10% fetal bovine serum, and 30% citrate buffer, pH7) and immediately transferred to a well from a ibidi µ-Slide 15 Well Glass Bottom slide. The cells were allowed to adhere for 15 min at room temperature, and the well was washed with 2% BSA solution in PBS before being fixed in 4% paraformaldehyde for 1 h at room temperature. The well was washed with PBS to remove the fixative solution and incubated for 20 min with 1:40 phalloidin Alexa Fluor 546 (1U, Thermo Fisher) and Hoechst 33342 (20 µM, Thermo Fisher) diluted in PBS. The incubation media was removed, the well washed with PBS, and a drop of ProLong Gold Antifade Mountant (Thermo Fisher) was added.

## In situ hybridization

Salivary glands (SGs) were dissected in PBS from uninfected or P. berghei-infected mosquitoes 21 days post-infection and immediately fixed in 4% paraformaldehyde for 1 h at room temperature. If tissues were not processed immediately, they were stored at 4 °C in paraformaldehyde until processing. We used the RNAscope™ Multiplex Fluorescent Assay (RNAscope Multiplex Fluorescent Reagent – ref# 323100 and RNAscope 4-plex Ancillary kit for multiplex fluorescent V2 – ref# 323120, Advanced Cell Diagnostics), employing three different spectral channels simultaneously. Briefly, SGs were washed with PBS to remove the fixative, then treated with 40 µL of protease IV (1 drop) at room temperature for 30 minutes to permeabilize the tissues. After two PBS washes (2 minutes each), SGs (10 pairs) were incubated with a probe mixture diluted 1:50. The following probes were used: Apyrase (Aga – AGAP011026 – C1, cat# 1302811-C1), PPO6 (Aga – PPO6 – C2, cat# 458541-C2), and Lipophorin (Aga – AGAP001826 – C3, cat# 484941-C3) (Advanced Cell Diagnostics). SGs were incubated with the probe mixture for 2 h with gentle shaking at 40 °C, then placed in 1 mL of 5x SSC and left overnight at room temperature. The next day, we developed the fluorescent signals for each probe following the manufacturer's instructions. For SG incubations (AMPs, HRP, and HRP blocker), 40 µL (1 drop) of reagent was used, and washes were performed with 1 mL of wash buffer (provided by the kit). The fluorescent signals were developed using Opal fluorophores (cat# NEL861001KT, Akoya Biosciences) as follows: C1 with Opal 520, C2 with Opal 570, C3 with Opal 620, and C4 with Opal 690. After the C4 signal was developed, tissues were counterstained with Hoechst (2 µM, Thermo Fisher) and Wheat Germ Agglutinin Alexa Fluor Plus 770 (2 µg/mL, Thermo Fisher) diluted in PBS for 20 min at room temperature. Finally, SGs were briefly rinsed with PBS and mounted with ProLong™ Gold Antifade Mountant (Thermo Fisher). Image analysis was performed on Imaris 9.9.1 (Bitplane, Concord, MA, USA) and Adobe Photoshop CC (Adobe Systems, San Jose, CA, USA).

## Western blot

SGs were dissected from uninfected, or *P. berghei*-infected mosquitoes 21 days post-infection and immediately placed in RIPA buffer (Thermo Fisher). A total of 10 pairs of SGs were then mixed with LDS sample buffer (Thermo Fisher) containing 5% β-mercaptoethanol (Sigma-Aldrich). Proteins were resolved on a NuPAGE™ 10% Bis-Tris Protein Gel (Invitrogen) under reducing conditions and transferred to a PVDF membrane using Invitrogen™ Power Blotter Select Transfer Stacks (Invitrogen). The membrane was blocked overnight at 4 °C with 5% milk powder in TBST (Tris-buffered saline with 0.1% Tween 20). For immunoblotting, membranes were incubated with primary antibodies: rabbit anti-lipophorin monospecific IgG (0.5 µ/ml) and mouse anti-Tubulin (1:1000, Thermo Fisher), diluted in TBST for 1 h at room

temperature. After washing three times with TBST, membranes were incubated with horseradish peroxidase-conjugated goat anti-mouse and goat anti-rabbit secondary antibodies (1:30,000 in TBST, Sigma-Aldrich) for 1 h at room temperature. Detection was performed using SuperSignal West Pico PLUS Chemiluminescent Substrate (Thermo Fisher Scientific).

For hemolymph samples, hemolymph was collected as previously described, and protein concentration was determined using Pierce™ BCA Protein Assay (Thermo Fisher). A total of 30 μg of hemolymph protein was mixed with reducing sample buffer, resolved on a NuPAGE™ 10% Bis-Tris Protein Gel, and transferred to a PVDF membrane as described above. Membranes were incubated overnight at 4 °C with rabbit anti-CLIPA14 (Bio Booster, 1:500) and mouse anti-apyrase (1:500) in blocking buffer. Following three washes with PBST, detection was conducted using SuperSignal West Pico PLUS Chemiluminescent Substrate for CLIPA14 and SuperSignal™ West Atto Ultimate Sensitivity Substrate for apyrase. Imaging was performed with an iBright Imager (Thermo Fisher).

### Real-time PCR
Pools of 20 pairs of SGs from uninfected or *P. berghei*-infected mosquitoes were collected into 200 μl of TRIzol™ reagent (ThermoFisher,) and homogenized using a motorized pestle. The SG homogenates were processed with TRIzol LS according to the manufacturer's protocol for RNA extraction. After RNA precipitation with isopropanol, the RNA pellets were washed twice with 300 μl of 75% ethanol. The tubes were mixed by vortexing and centrifuged at 7500 RCF for 5 min at 4 °C. The RNA pellets were then solubilized in 21 μl of RNase-free water by gentle pipetting, followed by incubation at 55 °C for 10 min to fully resuspend the RNA. All extracted RNA was used for complementary DNA (cDNA) synthesis using the QuantiTect Reverse Transcription Kit (Qiagen, Germantown, MD, USA) according to the manufacturer's instructions. Gene expression was assessed by quantitative PCR (qPCR) on a QuantStudio 7 real-time PCR system (Applied Biosystems). Semi-quantitative qPCR was used to measure the expression levels of the genes A5R1, Apy, AAPP, NT5E, D7R3, Tranf 1, LP, CLIPA14, and PPO6. Relative expression levels were normalized to *An. gambiae* ribosomal protein S7 (RpS7) as the internal control, and data analysis was conducted using the ΔΔ Ct method[72,73]. Statistical analysis of the fold change in gene expression was performed using an unpaired *t*-test (GraphPad, San Diego, CA, USA). Each experiment included at least three biological replicates per condition. The primers used for qPCR are listed in Supplementary Fig. 13.

### Ethical
This research complies with all relevant ethical regulations. The study was performed in strict accordance with the recommendations from the Guide for Care and Use of Laboratory Animals of the National Institutes of Health (NIH). The animal use was done in accordance with the Use Committee or The NIH Animal Ethics Proposal SOP LMVR 22.

### Reporting summary
Further information on research design is available in the Nature Portfolio Reporting Summary linked to this article.

## Data availability
All data needed to evaluate the conclusions are included in the paper. The mass spectrometry proteomics data have been deposited to the ProteomeXchange Consortium via the PRIDE partner repository with the dataset identifiers: PXD057628 (Salivary gland), 7PXD05758 (Saliva), and PXD057585 (Hemolymph). Source data are provided with this paper.

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

## Acknowledgements

The authors are grateful to Ryan Kissinger, Senior Medical Illustrator and Animator at NIAID, for the design of Fig. 9. We are grateful to Andre Laughinghouse and Kevin Lee for insectary support. We would like to thank Drs. Abhai Tripathi and Godfree Mlambo, and Chris Kizito the Insectary and Parasitology Core Facilities team at the Johns Hopkins Malaria Research Institute (JHMRI). We are grateful to Bloomberg Philanthropies for support of this facility and the work being performed at JHMRI. This work was supported by the NIH Distinguished Scholars Program and the Intramural Research Program of the Division of Intramural Research, NIAID, NIH AI001250-01 to J.V.R., Z01AI000947 to CBM, Z01 AI000810-21 to JMCR and Malaria Research Program Fellowship to ZRP. This work was supported by a JHMRI fellowship to S.K., the National Institutes of Health (R01AI132359 to P.S. and R03AI180804 to S.K.), the Howard Hughes Medical Institute to P.S. and S.K.; and the German Research Foundation grant SPP 2225 (FR2140/12-2) to F.F. B.L. was supported by an American Heart Association Postdoctoral Fellowship (2022-2024; 23POST1011626) and is currently supported by a University of Adelaide Future Making Fellowship and Australian Research Council Discovery Project (DP250102307).

## Author contributions

Conceptualization: T.L.A.S., and J.V.R. Methodology: T.L.A.S., S.K., P.S., A.B.B.F, C.S., B.L., O.A.C.T, B.N., M.Z., M.S., L.P.D, I.C., S.A., C.B.M., and J.V.R.; Investigation: T.L.A.S., S.K., A.B.B.F., C.S., B.L., O.A.C.T., J.O., B.M.N., Z.R.P., T.P., D.A.A., M.Z., M.S., L.P.D., F.F, I.C., S.A., J.M.C.R., and J.V.R.; Formal analysis: T.L.A.S., S.K., C.S., B.L., S.A., P.S., and J.V.R.; Writing – original draft: T.L.A.S. and J.V.R.; Supervision: J.V.R.

## Funding

## Competing interests

The authors declare no competing interests.

## Additional information

**Thiago Luiz Alves e Silva** [1] ✉, **Sachi Kanatani** [2], **Ana Beatriz Barletta Ferreira** [1], **Cindi L. Schwartz** [3], **Benjamin Liffner** [4,9,10], **Octavio A. C. Talyuli**[1], **Janet Olivas**[1,11], **Bianca M. Nagata**[5], **Zarna Rajeshkumar Pala**[1,12], **Tales Pascini**[1], **Derron A. Alves**[5], **Ming Zhao** [6], **Motoshi Suzuki** [6], **Lilian P. Dorner** [7], **Friedrich Frischknecht** [7,8], **Isabelle Coppens** [2], **Carolina Barillas-Mury** [1], **Sabrina Absalon** [4], **Jose M. C. Ribeiro**[1], **Photini Sinnis** [2] & **Joel Vega-Rodriguez** [1] ✉

[1]Laboratory of Malaria and Vector Research, National Institute of Allergy and Infectious Diseases, National Institutes of Health, Rockville, MD, USA. [2]The W. Harry Feinstone Department of Molecular Microbiology and Immunology and Johns Hopkins Malaria Research Institute, Bloomberg School of Public Health, Johns Hopkins University, Baltimore, MD, USA. [3]Microscopy Unit, Research Technologies Branch, Rocky Mountain Laboratories, National Institute of Allergy and Infectious Diseases, National Institutes of Health, Hamilton, MT, USA. [4]Department of Pharmacology & Toxicology, Indiana University School of Medicine, Indianapolis, IN, USA. [5]Infectious Disease Pathogenesis Section, National Institute of Allergy and Infectious Diseases, National Institutes of Health, Rockville,

MD, USA. ⁶Research Technologies Branch, National Institute of Allergy and Infectious Diseases, National Institutes of Health, Rockville, MD, USA. ⁷Integrative Parasitology, Center for Infectious Diseases, University of Heidelberg Medical School, Heidelberg, Germany. ⁸German Center for Infection Research, DZIF Partner Site Heidelberg, Heidelberg, Germany. ⁹Present address: School of Biological Sciences, University of Adelaide, Adelaide, SA, Australia. ¹⁰Present address: Institute for Photonics and Advanced Sensing, University of Adelaide, Adelaide, SA, Australia. ¹¹Present address: Department of Pathology, New York University Grossman School of Medicine, New York, NY, USA. ¹²Present address: Biological Sciences Graduate Program, University of Maryland, College Park, MD, USA. ✉e-mail: thiagoluiz.alvesesilva@nih.gov; joel.vega-rodriguez@nih.gov

