## [Peer Review file · Nature Communications]

High-Resolution Proteomics Unveils Salivary Gland Disruption and Saliva-Hemolymph Protein Exchange in *Plasmodium*-Infected Mosquitoes

Corresponding Author: Dr Joel Vega-Rodríguez

Version 0:

Reviewer comments:

Reviewer #1

(Remarks to the Author)

This manuscript, by the authors, reports a proteome analysis in *Plasmodium*-infected anopheline mosquitoes. The analysis shows that salivary glands infected with sporozoites are exchanging proteins between the hemolymph and salivary glands. I find this interesting.

Overall, the paper is well written. *Plasmodium* parasites are believed to traverse the SG cells when it infects them, and the epithelial cells of SGs are damaged at that time. Thus, at first glance, their results seem reasonable. The results may contribute to the elucidation of the mechanism of malaria transmission by mosquitoes and to malaria control. However, we believe that there are still some points to be discussed and modified in order to conclude this theory. If they are improved, the conclusion will be more solid.

It was unfortunate that the authors did not examine what effect the exchange of proteins between the SGs and hemolymph has on the malaria parasite. This makes it difficult to determine how much of an impact "protein exchange" has on the growth and transmission of malaria parasites. I believe that an analysis of this would be more valuable information.

Major comments:

- 1: Information on the uninfected mosquitoes used in the comparative analysis is unclear. Were the uninfected mosquitoes individuals that fed blood at the same time as the infected mosquitoes (dissected 21 days after blood feeding)? The presence or absence of blood feeding in the samples is important because blood feeding alters gene expression in the hemolymph and SGs, but this is not clearly stated in the text.
- 2: Some of the proteins derived from hemolymph detected in SGs are presumed to have been imported by adhesion to sporozoites rather than by diffusion (e.g., TEP15, PPO6). These should be discussed separately from the "protein exchange" proposed by the authors. Therefore, the humoral proteins detected in SGs should be examined for adhesion to sporozoites prior to entry into the SGs.
- 3: I think it should be analyzed for mosquitoes in the stage prior to sporozoite salivary gland invasion. If there is no protein exchange in infected mosquitoes prior to sporozoite salivary gland invasion, it would be stronger to say that sporozoite invasion is the trigger for "protein exchange".
- 4: Protein exchange between the SGs and hemolymph is thought to occur after the sporozoite enters the SGs. Since the life cycle of malaria parasites in the mosquito is almost complete at this stage, protein exchange may have little effect on parasite growth and infectivity. Is there any difference between sporozoites entering the SGs early (e.g., 14 days after infection or earlier) and sporozoites in the SGs after sufficient protein exchange has occurred? I believe that more analysis and discussion is needed on the period after infection with malaria. Instead, it is the physical condition of the mosquito that is affected by this. We believe that more consideration should be given to the impact on lifespan and fertility. On the other hand, we think it is easy to accept the theory that changes in the composition of saliva can alter blood-feeding behavior, such as probing, and thus affect the rate of transmission to the host.
- 5: Did the amount of protein exchange between SGs and hemolymph correlate with the number of sporozoite infections? The higher the number of infections, the more damage to the SGs and the more protein leakage and entry should occur. Also, is there a difference in protein exchange between LL and CL in the salivary glands? Most sporozoites are concentrated in LL; if they invade LL directly, then the amount of change in the constituent proteins would be different between LL and CL. Do you have any findings on these? I believe that the information would be very useful.
- 6: Fig3C, D: ISH analysis of apyrase shows that fluorescence is also detected in cavity. Does this mean that RNA is present

in saliva? If it is leaking from the cells, the sample handling conditions are a little questionable.

7: Was the presence of sporozoites confirmed in the cavity with altered saliva composition in Fig. 5F? If multiple sections are observed, some sporozoites should be visible, but this is not mentioned.

Minor comments:

1: In the list of proteins in Fig2D, shouldn't CLIPB3 be CLIPB4? Or is CLIPB4 missing from the figure?

2: Fig3E: What does the magenta in Panel2 indicate?

3: Line604: Looking at the figure, Pf infection results are for 3 assays, so the text should be corrected.

4: I think there is one author who are not listed in the authors' contributions section. Please check and correct it.

Reviewer #2

(Remarks to the Author)

In the manuscript, the authors provide an extended catalogue of proteins in mosquito salivary glands and in saliva upon infection by both *P. berghei* and *P. falciparum*. They also demonstrate that infection triggers damage to SG, facilitating protein exchange between SG and haemolymph. While the authors conduct multiple experiments including high throughput MS, immunostaining confirmation, porosity of SG through dextran beads, TEM and saliva proteomics, there is a lack of mechanistic characterization of the regulated proteins and the data are descriptive.

Please see a few minor comments below:

Introduction

There is information about protein contents, including proteins involved in feeding and in metabolism. One would expect that the study will determine the function of salivary proteins in these domains.

Results:

L. 362: there is no demonstration of epithelial disruption.

Minor comments:

L. 213: surprising to see a reference to fig 5B here.

L. 229: a word is missing, maybe "4.8 fold"

Reviewer #3

(Remarks to the Author)

Review of Manuscript

"High-Resolution Proteomics Unveils Salivary Gland Disruption and Saliva-Haemolymph Protein Exchange in Plasmodium-Infected Mosquitoes"

By Dr. Vega-Rodríguez et al.

Reviewer: Dr D. Leroy

Date: 24/01/2025

This manuscript provides a comprehensive proteomics analysis comparing the salivary glands and saliva of uninfected vs. Plasmodium-infected Anopheles mosquitoes. The authors catalog thousands of proteins, focusing on several key candidates through additional studies, such as immunofluorescence. They conclude that only a small percentage of proteins are differentially detected in the salivary glands and saliva of infected mosquitoes. Furthermore, certain proteins associated with sporozoite surface are depleted from the saliva of infected mosquitoes.

The manuscript is well-written and easy to follow. To my knowledge, this is the most extensive and in-depth proteomics study conducted on the salivary glands and saliva of malaria-transmitting mosquitoes, making a significant contribution to our understanding of the transmission of this parasitic disease. As such, I strongly support its publication in a high-impact journal like Nature Communications, provided the authors address the following points.

Minor Points

- Abstract: SG not explained.
- Line 397: "Infected mosquitoes contained 15,000 and 80,000 sporozoites" should read: "Infected mosquitoes contained between 15,000 and 80,000 sporozoites."

Major Points

Results:

- Line 130: The proteomics analysis of mosquito SGs was performed 21 days post-infection, but sporozoites can persist in mosquitoes for their entire lifespan (>50 days). Can the authors explain why they chose 21 days for the analysis and whether a different outcome might have been observed if the analysis had been conducted at a later time point? This is particularly relevant given that less than 2% of proteins are differentially detected in infected vs. uninfected mosquitoes, and proteins like EndoU are described in the literature as lipid homeostasis regulator during aging.

- Line 198: "Alpha-amylases are enzymes expressed in the anterior section of the salivary glands, a region not invaded by sporozoites." Could the authors elaborate on whether sporozoites might affect SG protein expression from a distance during infection? Could mediators be transported via the hemolymph?
- Line 226: Sentence should start with "In iSGs, ..."
- Lines 246-252: CLIPB4 is found to be 12 times more abundant in iSGs, while its negative regulators, CLIPA14 and CLIPA7, are upregulated and only detected in iSGs. Could the authors clarify whether these two proteins might counteract the immune response triggered by CLIPB4 during sporozoite infection? This section on immune response regulation is critical but complex, and the various hypotheses would benefit from being illustrated in a figure (scheme). This would help non-specialist readers better understand the intricate protein-mediated signaling regulating immune responses.
- Line 399: "We identified an average of 85 proteins in the saliva of uninfected *An. gambiae* mosquitoes and 41 in *P. berghei*-infected mosquitoes. Similarly, we identified an average of 106 proteins in uninfected saliva and 69 in *P. falciparum*-infected saliva..." Can the authors explain why data for uninfected mosquitoes are reported twice? Were the *P. berghei* and *P. falciparum* experiments performed on different days? If so, how reliable is protein detection across these time points (85 vs. 106 proteins)? Is the 20% variability typical for proteomics studies? How many proteins might be missed beyond 106, and can the authors roughly estimate this?
- Line 431: "The depletion of these proteins may facilitate sporozoite transmission by prolonging probing time and increasing the biting rate of infected mosquitoes." Could the authors clarify which molecular or biochemical pathways might support this hypothesis? For example, could a reduced immune response facilitate sporozoite survival and transmission? What mechanistic links could explain the relationship between protein depletion in saliva, prolonged probing time, and increased biting rate?
- Line 422: "Interestingly, a strong trend of protein downregulation was observed, with no proteins upregulated in saliva from iSGs. Analysis of the differentially expressed proteins revealed a notable trend: several proteins that were upregulated in the SGs following sporozoite invasion exhibited depletion in the saliva collected from infected mosquitoes." Could the authors clarify why they use terms like "upregulated" and "downregulated" when the observed differences might be due to protein transfer between compartments rather than differential expression? Could they demonstrate this with techniques like sporozoite purification and immunodetection of associated proteins (WB or immunoprecipitation)? I recommend using the term "level detected" rather than "expression" in line 452. Additionally, in Figure 8 (note: none of the figures are numbered in the PDF), the legend mentions "uninfected and *P. berghei*-iSGs," but where are the images of uninfected controls? Where is the AAPP control? Did the authors use blocking peptides or proteins to validate antibody specificity?

Discussion:

- Line 499: "Further studies are needed to determine whether sporozoites exploit host nutrients during the salivary gland stage." Could the authors briefly describe these potential studies? Functional studies investigating the role of host nutrients in sporozoite viability and how the parasite captures these nutrients would be valuable in identifying new therapeutic targets.
- Line 503: "This is important, as SGs may be contaminated with other tissues during dissection, such as fat bodies and hemocytes, which respond to infection." This is a key point for validating the entire study. How can the authors ensure that their protein identifications in the SGs, saliva, and sporozoite surfaces are not artifacts of the dissection procedure? Could they include additional controls?
- Line 505: "The fact that the hemolymph immune proteins upregulated in the proteome of iSGs were not upregulated in the RT-qPCR and in situ hybridization analysis suggests that these proteins most likely entered the SGs due to structural damage to the epithelial layer, as observed in our dextran diffusion assay and previously reported." The authors assume that only transcriptional regulation occurred. Could translational regulation also explain the findings? This hypothesis should be considered before concluding that leakage from the SGs during dissection is the sole explanation.
- Line 559: "This study offers a functional description of sporozoite salivary gland interactions..." While the authors have provided an extensive catalog of proteins involved, the functional relationships between these proteins remain underexplored. The authors should select key proteins for further functional studies and justify their choices. Additionally, could an AI/ML approach be applied to identify the most critical proteins for future investigation? How feasible is this method across different insect species, and could it help pinpoint proteins that play crucial roles in infection and immune regulation? A discussion on the potential of such approaches would enrich the manuscript and open the door for future studies in this area.

Reviewer #4

(Remarks to the Author)

In this manuscript, the authors examined the impact of *Plasmodium* infection on *Anopheles gambiae* salivary glands using proteomics, gene expression, and morphological analysis and the saliva proteome. The data provides new insights into the structural and cellular biology for mosquito salivary glands during sporozoite infection and may contribute to guiding the development of strategies to interrupt parasite transmission. The data presented here are very much worthy of publication. However, there are some details about the methodology and analysis missing need to be addressed. Below are some comments for authors to further improve the manuscript.

1. A flowchart of proteomics analyses in fig1 will help to understand the experimental design.
2. Line 574, in method, only GFP strain was used for proteomics analysis, but in Supplementary Figure 1: both GFP and mCherry-infected strains used for proteomics analysis.
3. All tables are not ready for publication. The tables should be labeled clearly with table title, column name explanations, et al.. Currently, we have to guess which one is which one.
4. Line 684: The detail method about how the differential expression (DE) cutoff was decided is not clear. In fig 2C, there are some DE proteins have very small fold changes and very small p value when n=3, it worths to check/explain.
5. Line 217: which is sup table 6 and which protein is mitochondrial pyruvate carrier? The same question for sup table 7?
6. Line 604: are the 50 mosquitoes the total for all groups or for each group?

Version 1:

Reviewer comments:

Reviewer #1

(Remarks to the Author)

The authors sincerely addressed the questions and concerns raised in my comments and revised the manuscript as appropriate. The additional data strengthened the authors' thesis. I am generally satisfied with the revised manuscript.

Minor comments:

- Line 472: Please correct "oocyts" to "oocyst."
- Supplement Figures 7 and 8: The numbers indicated by "scale bar" in the legends for each figure are "XX", please describe the correct numbers.
- Supplement Figure 7b: A larger image of "sporozoites lodged in the intercellular space" would help readers understand.

Reviewer #2

(Remarks to the Author)

The authors have generated new data using transgenic parasite that do not penetrate SG cells. These new data strongly reinforce their hypothesis, rendering the overall manuscript more convincing. They have also addressed the comments from reviewers. I recommend acceptance.

Reviewer #4

(Remarks to the Author)

The authors have addressed all my questions and my further comment:
Supp figure 1a: need to be polished, both 3 and 4 are "Protein elution/digestion"??

REVIEWER COMMENTS

Reviewer #1 (Remarks to the Author):

This manuscript, by the authors, reports a proteome analysis in
Plasmodium-infected anopheline mosquitoes. The analysis shows that
salivary glands infected with sporozoites are exchanging proteins
between the hemolymph and salivary glands. I find this interesting.

Overall, the paper is well written. Plasmodium parasites are believed to
traverse the SG cells when it infects them, and the epithelial cells of SGs
are damaged at that time. Thus, at first glance, their results seem
reasonable. The results may contribute to the elucidation of the
mechanism of malaria transmission by mosquitoes and to malaria
control. However, we believe that there are still some points to be
discussed and modified in order to conclude this theory. If they are
improved, the conclusion will be more solid.

It was unfortunate that the authors did not examine what effect the
exchange of proteins between the SGs and hemolymph has on the
malaria parasite.

This makes it difficult to determine how much of an impact “protein
exchange” has on the growth and transmission of malaria parasites. I
believe that an analysis of this would be more valuable information.

Major comments:

1: Information on the uninfected mosquitoes used in the comparative
analysis is unclear. Were the uninfected mosquitoes individuals that fed
blood at the same time as the infected mosquitoes (dissected 21 days
after blood feeding)?

The presence or absence of blood feeding in the samples is important
because blood feeding alters gene expression in the hemolymph and
SGs, but this is not clearly stated in the text.

Yes, uninfected mosquitoes obtained a blood meal at the same time as
infected mosquitoes. We have edited the Results (Lines 134-135) and
M&M (Lines 590-592) to clarify this.

2: Some of the proteins derived from hemolymph detected in SGs are
presumed to have been imported by adhesion to sporozoites rather than
by diffusion (e.g., TEP15, PPO6). These should be discussed separately
from the “protein exchange” proposed by the authors. Therefore, the
humoral proteins detected in SGs should be examined for adhesion to
sporozoites prior to entry into the SGs.

We appreciate the reviewer's suggestion to distinguish between proteins
entering the salivary glands via diffusion and those transported by
adhesion to sporozoites. In addition to diffusion, we recognize that some
hemolymph proteins may be introduced into the salivary glands while
bound to sporozoites. To address this, we have incorporated new
immunofluorescence data addressing the binding of specific hemolymph
proteins to sporozoites (Supplementary Figure 11). We observe binding
of some of these proteins to hemolymph sporozoites, but the binding
seems much weaker than that detected within the salivary glands. For
example, our data show that lipophorin, an abundant hemolymph protein
in infected salivary glands, does not associate with hemolymph
sporozoites (Supplementary Figure 11), despite being highly abundant
and having a widespread and diffuse distribution within the infected
salivary glands (Supplementary Figure 2). Given lipophorin's high
molecular weight (366 kDa) and the enrichment of the most abundant
hemolymph proteins within the infected glands, it is reasonable to infer
that diffusion is the primary mechanism of entry (Supplementary Figure
5).

Further evidence comes from our dextran experiments, where we
observed dextran diffusion into infected salivary glands 21 days post-
infection, when most of the invasion had already happened (Figure 5d,
e). Similarly, we find that the most abundant salivary proteins are
present in the hemolymph of infected mosquitoes, suggesting diffusion

out of the salivary glands, opposite to the direction of sporozoite
invasion, as confirmed by Western blot and hemolymph proteomics
(Figure 5c, Supplementary Figure 5).

In addition, we provide new data showing that the membrane contact
between salivary gland cells is disrupted in infected salivary glands
(Figure 5i, Supplementary Figure 6d), and the structure of the secretory
cavities is severely compromised in highly infected glands
(Supplementary Figure 7c and 8a).

Altogether, we propose that salivary gland damage by sporozoite
invasion is the main driver for the bidirectional protein diffusion, while
acknowledging that some hemolymph proteins may enter the glands
while bound to sporozoites. We have now expanded the discussion to
include the possibility that certain hemolymph proteins are transported
into the salivary glands via sporozoite adhesion.

3: I think it should be analyzed for mosquitoes in the stage prior to
sporozoite salivary gland invasion. If there is no protein exchange in
infected mosquitoes prior to sporozoite salivary gland invasion, it would
be stronger to say that sporozoite invasion is the trigger for “protein
exchange.

Continuing the discussion from the previous point, we referenced in the
manuscript the results by Wells and Andrew (2019, PMID: 31387905)

that demonstrated that infected glands exhibit cytoplasmic disruption,
basal accumulation of saliva, and extensive damage, reducing secretory
protein abundance. Importantly, they showed that this damage correlates
with parasite invasion rather than parasite load, a finding consistent with
our dextran assays (New Supplementary Figure 9b). Furthermore, we
performed new dextran experiments with mosquitoes infected 9 days
post blood feeding, when the sporozoites are still inside the oocysts, and
show that dextran uptake between uninfected and infected salivary
glands is similar and in a small proportion of the glands (New
Supplementary Figure 9a). Our study provides additional quantitative
and qualitative evidence of glandular disruption and the resulting
exchange of saliva and hemolymph proteins.

We also added a new set of experiments using expansion microscopy to
demonstrate that successful invasion of the salivary gland by sporozoites
is required to trigger the changes in saliva appearance observed in our
transmission electron microscopy images (Supplementary Figures 7 and
8). For these experiments, we used a transgenic *P. berghei* line with
conditional knockdown of the rhoptry protein RON11 (RON11cKD).
These sporozoites are unable to form the tight junctions typically seen
during salivary gland invasion and therefore attempt to invade the
junctional regions of the gland but fail to enter the salivary cavities. Our
results show that only successful sporozoite invasion leads to the
observed morphological changes in the gland.

In addition, we used expansion microscopy to quantify these alterations,
providing measurements of saliva pattern changes, gland integrity, and
saliva density (Supplementary Figures 7 and 8). Saliva density was
quantified using Fourier Transform (FT) analysis of images from
secretory cavities. FT decomposes pixel patterns into frequency
components, enabling the objective measurement of texture and
structural features within the cavities.

Our original manuscript included multiple assays supporting that protein
exchange is triggered by sporozoite invasion. For example, in Figure 5B,
Western blot analysis shows that lipophorin is detected in the salivary
glands at 21 days post-feeding, but only in infected mosquitoes.

Similarly, Figure 5C demonstrates that salivary apyrase is absent from
the hemolymph at 9 days post-blood feeding—before sporozoite
invasion—but is detected at 21 days in infected mosquitoes, coinciding
with sporozoite invasion of the salivary glands.

Additionally, our hemolymph proteomic analysis at 19 days post-feeding
detected salivary gland proteins only in the hemolymph of infected
mosquitoes.

These findings reinforce our conclusion that protein exchange is a
consequence of sporozoite invasion of the salivary glands.

4: Protein exchange between the SGs and hemolymph is thought to
occur after the sporozoite enters the SGs. Since the life cycle of malaria
parasites in the mosquito is almost complete at this stage, protein
exchange may have little effect on parasite growth and infectivity. Is
there any difference between sporozoites entering the SGs early (e.g., 14
135 days after infection or earlier) and sporozoites in the SGs after sufficient
protein exchange has occurred? I believe that more analysis and
discussion is needed on the period after infection with malaria. Instead,
it is the physical condition of the mosquito that is affected by this. We
believe that more consideration should be given to the impact on
lifespan and fertility. On the other hand, we think it is easy to accept the
theory that changes in the composition of saliva can alter blood-feeding
behavior, such as probing, and thus affect the rate of transmission to the
host.

We appreciate the reviewer's insightful comment regarding the impact of
protein exchange between the salivary glands and hemolymph on
malaria parasite viability. While this is indeed an important question,
investigating it presents significant challenges due to the complexity of
the experiments and the difficulty in controlling relevant variables.
Parasite viability could be assessed in the hemolymph, salivary glands,
or during salivation by measuring parasite motility, viability, or
infectivity. However, our findings indicate alterations in multiple
salivary and hemolymph proteins, making it challenging to replicate

these changes specifically in the salivary glands. To address this, we are
currently developing new tools for tissue-specific gene expression
regulation in mosquitoes, which in the future, will enable us to further
explore the potential effects of these proteins on parasite viability.
However, we feel that it is important to point out the changes in salivary
gland proteins and how the function of these proteins could potentially
affect parasite viability, based on the current knowledge from other
studies about the biology of sporozoite transmission. This is discussed
throughout the manuscript.

The impact of protein exchange on mosquito lifespan and fertility is an
important question. However, as explained above, current tools do not
allow us to isolate the effects of protein exchange from those induced by
the parasite itself. Numerous studies have shown that parasite infection
influences mosquito lifespan and fertility (Vézilier et al., 2012, PMID:
22859589; Hurd and Hug, 1995, PMID: 8559587; Jahan and Hurd,
1997, PMID: 9290843; Ferguson et al., 2003, PMID: 14761058; Ahmed
and Hurd, 2006, PMID: 16213176; Werling et al., PMID: 16213176).

These effects depend on multiple factors, including the mosquito-
parasite combination, infection intensity, and number of blood meals.
Given these complexities, current methodologies make it difficult to
determine how parasite-induced changes in hemolymph and saliva
protein composition due to salivary gland invasion affect mosquito
survival and fertility. However, the system we are developing for tissue-

specific gene regulation will enable more precise investigations by
allowing targeted control of multiple genes. We have expanded the
discussion to emphasize the relevance of future studies on the potential
effect on mosquito fitness and fertility.

We also agree that alterations in saliva composition influence mosquito
feeding behavior and parasite transmission. As discussed in the
manuscript, reduced levels of salivary gland proteins can enhance
mosquito probing activity, increasing the likelihood of sporozoite
inoculation into the host dermis. For example, previous studies have
shown that decreased levels of salivary proteins, such as aegyptin and
apyrase, can prolong probing time (Campos Chagas et al., 2014, PMID:
24778255; Ribeiro et al., 1985, Journal of Insect Physiology 31:689).

We have expanded the discussion to propose how the hemolymph could
be “clotting” or changing the physical properties of the saliva in the
regions invaded by the parasite, and its potential effect on increasing the
mosquito probing time and parasite transmission. This hypothesis was
first proposed in 1984, by Rossignol, Ribeiro and Spielman (PMID:
6696175), and our data nicely support their hypothesis.

5: Did the amount of protein exchange between SGs and hemolymph
correlate with the number of sporozoite infections? The higher the
number of infections, the more damage to the SGs and the more protein
leakage and entry should occur

The new expansion microscopy data shows that salivary glands with a
high level of infection exhibit more severe damage, as evidenced by the
loss of salivary cavity definition (Supplementary Figs 7 and 8).

However, there is no correlation between the number of sporozoites in
the salivary gland and dextran uptake or changes in saliva appearance,
such as granularity or "clotting." We have now included additional
images from the dextran experiment, showing that heavily infected
salivary glands can display a lower dextran signal compared to glands
with fewer sporozoites but a stronger dextran signal (Supplementary
Figure 9b). The consistent finding is that dextran infiltrates infected
salivary glands regardless of parasite numbers. Similarly, the granularity
of saliva in invaded cavities occurs irrespective of whether they contain
a low or high number of sporozoites. A similar phenomenon was
reported by Wells and Andrew (2019, PMID: 31387905).

Also, is there a difference in protein exchange between LL and CL in the
salivary glands? Most sporozoites are concentrated in LL; if they invade
LL directly, then the amount of change in the constituent proteins would
be different between LL and CL. Do you have any findings on these? I
believe that the information would be very useful.

The sporozoites primarily invade the lateral lobes (LL). As shown in
Figure 3A–C, hemolymph proteins detected in the salivary glands
exhibit stronger staining intensity in the LL, where parasites are present,
as indicated by anti-CS antibody staining. However, some medial lobes

also show staining for certain hemolymph proteins, suggesting that
diffusion to the medial lobe may occur.

Regarding the diffusion of salivary proteins between lobes, staining for
AAPP in Figure 3D, which is a protein known to be expressed in the LL,
indicates that this process takes place, as evidenced by the weaker but
detectable AAPP signal in the medial lobe. Additionally, parasite
infection appears to induce the expression of salivary proteins in new
regions. For example, apyrase mRNA, which is normally expressed only
in the medial lobe of non-infected glands, is also detected in the lateral
lobes following sporozoite invasion (Figure 4C).

6: Fig4C, D: FISH analysis of apyrase shows that fluorescence is also
detected in cavity. Does this mean that RNA is present in saliva? If it is
leaking from the cells, the sample handling conditions are a little
questionable.

This result is reproducible and specific to the apyrase probe in the
central lobe. We do not believe it is an artifact or mechanical damage to
the gland, as no issues were observed with other probes that we have
tried in our laboratory. Extracellular RNAs, including microRNAs, have
been detected in mosquito saliva (Maharaj et al., 2015, PMID:
25612225; Gold et al., 2020, PMID: 32927629; Fiorillo et al., 2022,
PMID: 35681077; Yeh et al., 2023, PMID: 36996041), and dsRNA has
been reported in various insects (Santos et al., 2021, PMID: 33806650).

Thus, the presence of apyrase mRNA in mosquito saliva, while
intriguing, is not difficult to reconcile with the presence of extracellular
RNAs. Its potential role in salivary gland biology and pathogen
transmission will be explored in future studies. We have added
additional text to the Results section to point out this result.

7: Was the presence of sporozoites confirmed in the cavity with altered
saliva composition in Fig. 5F? If multiple sections are observed, some
sporozoites should be visible, but this is not mentioned.

Yes, it was confirmed. We have delineated the regions containing
sporozoites in cavities with altered saliva composition on Figure 5F.

Minor comments:

1: In the list of proteins in Fig2D, shouldn't CLIPB3 be CLIPB4? Or is
CLIPB4 missing from the figure?

CLIPB3 changed to CLIPB4.

2: Fig3E: What does the magenta in Panel2 indicate?

The magenta represents WGA, which stains the salivary gland surface.
This was noted in the figure legend, and we have now labeled "WGA" in
the figure.

3: Line604: Looking at the figure, Pf infection results are for 3 assays, so
the text should be corrected.

This has been corrected in the figure legend.

4: I think there is one author who are not listed in the authors'
contributions section. Please check and correct it.

Corrected

Reviewer #2 (Remarks to the Author):

In the manuscript, the authors provide an extended catalogue of proteins
in mosquito salivary glands and in saliva upon infection by both P.
berghei and P. falciparum. They also demonstrate that infection triggers
damage to SG, facilitating protein exchange between SG and
haemolymph. While the authors conduct multiple experiments including
high throughput MS, immunostaining confirmation, porosity of SG
through dextran beads, TEM and saliva proteomics, there is a lack of
mechanistic characterization of the regulated proteins and the data are
descriptive.

While we agree that some of our proteomic data is descriptive, we
provide clear evidence of the mechanism by which hemolymph proteins
enter the salivary glands following sporozoite invasion and how salivary
proteins leak into the hemolymph. Previous transcriptomic and
proteomic studies suggested that sporozoite infection induces an

immune response in the salivary glands by upregulating immune genes
in that tissue. However, our findings demonstrate that the immune
proteins detected in infected salivary glands originate from the
hemolymph rather than being synthesized within the tissue. In addition,
the change in protein composition in both fluids, hemolymph and saliva,
could have significant implications for both, the parasite and the
mosquito. For example, reduction in some salivary proteins has been
shown to affect mosquito feeding (Campos Chagas et al., 2014, PMID:
24778255; Ribeiro et al., 1985, Journal of Insect Physiology 31:689).

In this revised manuscript, we contextualize our results with studies
indicating that infected mosquitoes have longer probing times and
reduced saliva apyrase activity (Rossignol, Ribeiro, and Spielman, 1984,
PMID: 6696175; Thiévent et al., 2019, PMID: 30986373). The study by
Rossignol et al., 1984, proposed that reduced apyrase activity results in
biochemical damage to specific gland regions (those invaded by the
sporozoites). We demonstrate that sporozoites cause physical damage,
altering saliva composition and appearance, and hypothesize that this
damage may enhance sporozoite transmission by reducing the salivary
gland's functional capacity, thereby increasing probing during biting and
enhancing sporozoite delivery. This is in agreement with previous work
and is now discussed in the Discussion section.

As mentioned in our response to Reviewer 1, determining the functional
role of this protein exchange in mosquito or parasite biology and
transmission requires a complex experimental approach. This would
necessitate the development of novel tools to achieve tissue-specific
regulation of selected proteins at precise stages of parasite development,
particularly during sporozoite invasion of the salivary glands.

Please see a few minor comments below:

Introduction

There is information about protein contents, including proteins involved
in feeding and in metabolism. One would expect that the study will
determine the function of salivary proteins in these domains.

As noted above, addressing this question is quite difficult and would
require the development of tools for tissue-specific regulation of selected
proteins at precise timepoint during parasite development, particularly
during sporozoite invasion of the salivary glands. However, our findings
indicate an overall decrease in salivary protein output. Based on
previous studies, we propose that this reduction may explain the
increased probing time and bite frequency observed in infected
mosquitoes. This, in turn, could enhance parasite transmission by
increasing the likelihood of sporozoite injection into the host dermis
during probing.

Results:

331 L. 362: there is no demonstration of epithelial disruption.

We have now included new data demonstrating significant ultrastructural
changes in the salivary gland induced by sporozoite invasion. Electron
microscopy reveals that in infected salivary glands, intercellular contact
between adjacent membranes is disrupted, forming a noticeable
intercellular cleft, whereas in uninfected glands, adjacent cell
membranes remain tightly abutted (Fig. 5i and Supplementary Fig. 6d).
Additionally, using expansion microscopy, we have observed and
quantified structural damage to salivary cavities caused by sporozoite
invasion. This damage is characterized by the loss of cavity definition,
marked by the disruption or shortening of the typical lateral extensions
that define the secretory cavity of salivary gland cells (Supplementary
Figures 7c, 8a, and 12). These experiments include a control with
RON11^{cKD} parasites (Bantuchai et al., 2019, PMID: 31247198), which
are deficient in the RON11 rhoptry protein and develop sporozoites
normally, but the majority (>99%) fail to successfully invade the
salivary gland. Rather, they try to enter the gland through the
intercellular space, where they remained trapped. Salivary glands from
RON11^{cKD} infected mosquitoes showed reduced damage to the secretory
cavities. These new findings, along with our original data from

proteomic analysis, Western blotting, and dextran diffusion assays,
indicate an indiscriminate exchange of saliva and hemolymph in regions
invaded by parasites, resulting from the destabilization of the gland
structure.

Minor comments:

356 L. 213: surprising to see a reference to fig 5B here.

It is not ideal, but we think that the figure 5B should remain within
Figure 5, validating epithelial integrity result.

359 L. 229: a word is missing, maybe “4.8 fold”

Added “times”

Transferrin was 4.8 times more abundant in iSGs.

Reviewer #3 (Remarks to the Author):

Review of Manuscript

“High-Resolution Proteomics Unveils Salivary Gland Disruption and
Saliva-Haemolymph Protein Exchange in Plasmodium-Infected
Mosquitoes”

By Dr. Vega-Rodríguez et al.

Reviewer: Dr D. Leroy

Date: 24/01/2025

This manuscript provides a comprehensive proteomics analysis
comparing the salivary glands and saliva of uninfected vs. Plasmodium-
infected Anopheles mosquitoes. The authors catalog thousands of
proteins, focusing on several key candidates through additional studies,
such as immunofluorescence. They conclude that only a small
percentage of proteins are differentially detected in the salivary glands
and saliva of infected mosquitoes. Furthermore, certain proteins
associated with sporozoite surface are depleted from the saliva of
infected mosquitoes.

The manuscript is well-written and easy to follow. To my knowledge,
this is the most extensive and in-depth proteomics study conducted on
the salivary glands and saliva of malaria-transmitting mosquitoes,
making a significant contribution to our understanding of the
transmission of this parasitic disease. As such, I strongly support its

publication in a high-impact journal like Nature Communications,
provided the authors address the following points.

Minor Points

• Abstract: SG not explained.

**Corrected.**

• Line 397: “Infected mosquitoes contained 15,000 and 80,000
sporozoites” should read:

“Infected mosquitoes contained between 15,000 and 80,000
sporozoites.”

**Corrected.**

Major Points

Results:

• Line 130: The proteomics analysis of mosquito SGs was performed 21
407 days post-infection, but sporozoites can persist in mosquitoes for their
entire lifespan (>50 days). Can the authors explain why they chose 21
409 days for the analysis and whether a different outcome might have been
observed if the analysis had been conducted at a later time point? This is

particularly relevant given that less than 2% of proteins are differentially
detected in infected vs. uninfected mosquitoes, and proteins like EndoU
are described in the literature as lipid homeostasis regulator during
aging.

Indeed, we acknowledge that these proteins may play a role in regulating
mosquito metabolism and mobilizing nutrients from the hemolymph. A
statement highlighting this possibility has been added to the discussion.

We selected 21 days post-blood feeding for our analysis because this
time point is commonly used in transmission studies (clarified now in
Lines 132-134 of the Results section). By this stage, the majority of
sporozoites have invaded the salivary glands, the peak of invasion has
passed, and the number of parasites circulating in the hemolymph has
significantly decreased. While sporozoites can be detected in the salivary
glands as early as 11–12 days post-infection and persist for an extended
period, previous studies provide insight into the temporal dynamics of
sporozoite infectivity. For example, Flores-Garcia et al. (2019, PMID:
31849326) demonstrated that the infectivity of *P. berghei* sporozoites
expressing *P. falciparum* CSP remained stable between days 18 and 28
post-infection. However, van Schuijlenburg et al. (2024, PMID:
38641838) reported differences in motility and *in vitro* infectivity of *P.*
*falciparum* sporozoites between 14 and 20 days post-infection.

Additionally, Porter et al. (1954, PMID: 13161969) found that *P.*
*gallinaceum* sporozoites persisted in *Aedes* mosquitoes for up to six

434 weeks, with a gradual decline in both sporozoite numbers and infectivity
over time.

Further studies are needed to determine how these proteins influence
parasite infectivity over time. Given the potential variability in outcomes
depending on the parasite-mosquito combination and whether infectivity
is assessed *in vivo* or *in vitro*, future work will incorporate precise
temporal control of gene expression during sporozoite invasion of SGs
to elucidate these effects under controlled conditions.

• Line 198: “Alpha-amylases are enzymes expressed in the anterior
section of the salivary glands, a region not invaded by sporozoites.”
Could the authors elaborate on whether sporozoites might affect SG
protein expression from a distance during infection? Could mediators be
transported via the hemolymph?

These are intriguing questions. It is certainly possible that sporozoites
could influence gene expression in the salivary gland from a distance.
We envision two potential mechanisms: 1) sporozoites invading the
distal lateral lobes might alter the expression in other gland regions, or
2) changes in salivary gland expression might be driven by the oocyst
stage, either before sporozoites are released or during their release. Our
laboratory is actively investigating these possibilities through a separate

project that characterizes salivary gland expression at various time
points—before, during, and after sporozoite invasion.

Some mediators could be imported with the hemolymph since the
invasion of hemolymph proteins into the gland seems to not be protein
specific. Therefore, any mediator present in the hemolymph at the
moment of invasion could enter the invaded portions of the salivary
gland.

We have significantly edited this section to better summarize the main
findings.

• Line 226: Sentence should start with “In iSGs, ...”

Corrected.

• Lines 246-252: CLIPB4 is found to be 12 times more abundant in
iSGs, while its negative regulators, CLIPA14 and CLIPA7, are
upregulated and only detected in iSGs. Could the authors clarify whether
these two proteins might counteract the immune response triggered by
CLIPB4 during sporozoite infection?

The role of CLIPB4 in sporozoite melanization has not been studied.

However, recent findings indicate that CLIPB4 is a central protease
directly activating PPOs (Zhang et al., 2023, PMID: 37461554).

Additionally, CLIPA14 has been shown to inhibit the melanization of
late-stage oocysts and sporozoites, serving a protective role (Zeineddine
et al., 2024, PMID: 39149419). Notably, CLIPA14 silencing is

incomplete, with significant protein levels persisting in the hemolymph,
which the authors of the study suggest may explain why silencing alone
does not fully eliminate the parasite. A knockout model, or a tissue-
specific knockdown system, would be more effective approaches to
studying their function on sporozoite biology in the salivary gland.

We have revised the discussion to highlight the potential negative
regulatory effects of CLIPA14 and CLIPA7 on CLIPB4: “*Thus, the*
*parasite may recruit negative regulators like CLIPA14 and CLIPA7 to*
*prevent PPO activation by agonists like CLIPB4.*”

This section on immune response regulation is critical but complex, and
the various hypotheses would benefit from being illustrated in a figure
(scheme). This would help non-specialist readers better understand the
intricate protein-mediated signaling regulating immune responses.

• Line 399: “We identified an average of 85 proteins in the saliva of
uninfected *An. gambiae* mosquitoes and 41 in *P. berghei*-infected
mosquitoes. Similarly, we identified an average of 106 proteins in
uninfected saliva and 69 in *P. falciparum*-infected saliva...” Can the
authors explain why data for uninfected mosquitoes are reported twice?

We have significantly edited this section to better summarize the main
findings. In addition, we have added a new figure scheme

(Supplementary figure 3c) to facilitate the reader better understand these
immune responses.

These experiments were conducted separately, with *P. berghei* infections
performed at the NIH and *P. falciparum* infections at Johns Hopkins
University. Since the Mass Spectrometry analyses were carried out at
different times, it was crucial to include a control group for each
experiment to minimize batch effects and ensure reliable results.

Were the *P. berghei* and *P. falciparum* experiments performed on
different days? If so, how reliable is protein detection across these time
points (85 vs. 106 proteins)? Is the 20% variability typical for
proteomics studies? How many proteins might be missed beyond 106,
and can the authors roughly estimate this?

In proteomic studies using label-free quantification, variability between
different batches is often seen. This variability arises from different
factors related to sample preparation, data acquisition and instrument
performance. Besides this, one challenge with this experiment is that the
saliva protein concentration is highly diluted, adding an additional layer
of variability. We are confident with these results as the same pattern of
changes in the saliva was seen for both parasite species.

• Line 431: “The depletion of these proteins may facilitate sporozoite
transmission by prolonging probing time and increasing the biting rate
of infected mosquitoes.” Could the authors clarify which molecular or
biochemical pathways might support this hypothesis? For example,
could a reduced immune response facilitate sporozoite survival and
transmission? What mechanistic links could explain the relationship
between protein depletion in saliva, prolonged probing time, and
increased biting rate?

We have now expanded the discussion to contextualize our results with
studies indicating that infected mosquitoes have longer probing times
and reduced salivary apyrase activity (Rossignol, Ribeiro, and Spielman,
1984, PMID: 6696175; Thiévent et al., 2019, PMID: 30986373). The
study by Rossignol et al., 1984, proposed that reduced apyrase activity
results in biochemical damage to specific gland regions (those invaded
by the sporozoites). We added new data demonstrating that sporozoites
cause physical damage with loss of cavity definition, marked by the
disruption or shortening of the typical lateral extensions that define the
secretory cavity of salivary gland cells (Supplementary Figures 7c, 8a,
and 12). These changes alter saliva composition and appearance, and
hypothesize that this damage may enhance sporozoite transmission by
reducing the salivary gland's functional capacity, thereby increasing
probing during biting and enhancing sporozoite delivery. This is in

agreement with previous work showing the reduction in apyrase activity
and increase in probing, and is now discussed in the Discussion section.

In addition, the Discussion of the original submission included this
statement:

*“Notably, for both parasites, we observed a reduction in yellow proteins*
*and EndoU and, for P. falciparum, a decrease in one D7 protein. All*
*other saliva proteins tended toward depletion, including apyrase. The*
*overall reduction in salivary proteins may impact parasite transmission*
*dynamics by increasing probing time and bite frequency, similar to the*
*effects observed when apyrase and D7 levels are reduced, ultimately*
*facilitating sporozoite transmission.”*

• Line 422: “Interestingly, a strong trend of protein downregulation was
observed, with no proteins upregulated in saliva from iSGs. Analysis of
the differentially expressed proteins revealed a notable trend: several
proteins that were upregulated in the SGs following sporozoite invasion
exhibited depletion in the saliva collected from infected mosquitoes.”

Could the authors clarify why they use terms like "upregulated" and
"downregulated" when the observed differences might be due to protein
transfer between compartments rather than differential expression?

We appreciate the reviewer's feedback and have revised the manuscript
accordingly:

We retained the terms "upregulated" and "downregulated" early in the
manuscript until we established that the salivary glands do not transcribe
these genes. From that point forward, we describe the changes as
"enrichment" or "depletion."

The text has been updated to reflect this distinction. *“These results*
*support that lipophorin, PPO6, and presumably other humoral proteins*
*enriched in iSGs, originate from external tissues. Henceforth, we will*
*refer to these proteins as enriched or depleted instead of up or*
*downregulated.”*

Could they demonstrate this with techniques like sporozoite purification
and immunodetection of associated proteins (WB or
immunoprecipitation)?

We have now included immunofluorescence data for sporozoites isolated
from mosquito hemolymph and infected salivary glands. In hemolymph
sporozoites, transferrin 1 was found to bind strongly to the sporozoites,
while lipophorin, PPO6, and CLIP14A showed weak binding, and no
binding was detected for TEP15 and vitellogenin (Supplementary Fig.
11a). All these proteins were enriched in the proteome of infected
salivary glands, with lipophorin being particularly abundant (Figure S2).
Therefore, it is unlikely that all the lipophorin observed in infected
glands enters bound to the sporozoite. These findings support the idea
that hemolymph proteins primarily enter infected salivary glands

through damaged epithelial barriers, rather than by associating with the
sporozoite surface.

For sporozoites isolated from infected salivary glands, only PPO6
showed faint residual binding. Transferrin 1 and TEP15 dissociated from
the sporozoite surface once the parasites were removed from the salivary
glands (Supplementary Fig. 11b). These results suggest that these
proteins are loosely bound to sporozoites or accumulate around them
only while they are within the salivary gland cavities. We currently do
not understand the mechanism by which these proteins localize around
the parasite or why they dissociate. One possibility is that changes in the
physical properties of the hemolymph influence this phenomenon, which
will be explored in follow-up studies.

I recommend using the term "level detected" rather than "expression" in
line 452.

Replaced “expression” for “detection”

Additionally, in Figure 8 (note: none of the figures are numbered in the
PDF), the legend mentions "uninfected and P. berghei-iSGs," but where
are the images of uninfected controls? Where is the AAPP control? Did

the authors use blocking peptides or proteins to validate antibody
specificity?

We have revised the figure legend to remove "uninfected," as this figure
includes only infected salivary glands. The control for uninfected glands
showing AAPP staining is provided in Figure S9.

The specificity of the PPO6 antibody was previously demonstrated by
Kwon and Smith (2019, PMID: 31235594). For AAPP, whole infected
and uninfected salivary gland images are shown in Figure 3D, where
AAPP expression follows the expected staining pattern in the lateral
lobes, consistent with previous studies (Wells and Andrew, 2015, PMID:
26627194; Wells et al., 2017, PMID: 28377572; Yoshida and Watanabe,
2006, PMID: 16907827; Klug et al., 2022, PMID: 36223382).

Additionally, the AAPP antibodies were raised against a specific AAPP-
derived peptide, and mono-specific IgG antibodies used for
immunofluorescence were purified using this peptide. We have updated
the Materials and Methods section to clarify this.

Discussion:

• Line 499: "Further studies are needed to determine whether sporozoites
exploit host nutrients during the salivary gland stage." Could the authors
briefly describe these potential studies? Functional studies investigating
the role of host nutrients in sporozoite viability and how the parasite

captures these nutrients would be valuable in identifying new therapeutic
targets.

We agree that these functional studies are valuable for identifying
potential targets or pathways for transmission-blocking interventions.

While we acknowledge the challenges associated with these assays in
the discussion of previous reviewer comments, we have included a brief
discussion of potential studies in the manuscript.

*“Further studies are needed to determine whether sporozoites exploit*
*host nutrients during the salivary gland stage or if hemolymph proteins*
*enriched in the salivary glands regulate nutrient mobilization.*

*Addressing this question will require tissue-specific downregulation of*
*key metabolic pathways during sporozoite invasion, followed by*
*functional assays to assess parasite viability. Emerging technologies are*
*being developed to enable such investigations.”*

• Line 503: “This is important, as SGs may be contaminated with other
tissues during dissection, such as fat bodies and hemocytes, which
respond to infection.” This is a key point for validating the entire study.
How can the authors ensure that their protein identifications in the SGs,
saliva, and sporozoite surfaces are not artifacts of the dissection
procedure? Could they include additional controls?

We acknowledge the concern regarding potential contamination from fat
body tissue and hemocytes during salivary gland dissections. To
minimize this, we carefully removed fat body tissue during sample
preparation, and microscopy analyses confirm that contamination is
minimal. However, given the sensitivity of proteomics and RNA
sequencing, trace detection of surrounding tissues is unavoidable.

To validate our findings, we performed multiple experiments, including
immunofluorescence assays (IFAs) and in situ hybridization, which
confirm that fat body cells and hemocytes remain external to the salivary
gland cavities, where saliva and sporozoites accumulate. Using tissue-
specific markers—lipophorin for fat body and PPO6 for hemocytes—we
demonstrate that these cells do not infiltrate the salivary glands. In
contrast, the proteins we investigated are localized within the salivary
gland cavities, as confirmed by IFAs, confocal microscopy of whole
glands, histological sections, and in situ hybridization.

Additionally, the proteins enriched within the infected salivary glands
are the most abundant hemolymph proteins, as we confirmed with the
hemolymph proteome. Our dextran assay further suggests that these
proteins enter the salivary glands due to tissue damage. Additionally,
new transmission electron microscopy data (Figure 5i and
Supplementary Figure 6d) show that sporozoite infection destabilizes the

salivary gland septate junctions, creating intercellular spaces that may
facilitate bidirectional protein movement.

Together, these findings strongly indicate that contamination from
external tissues does not account for our results, reinforcing the validity
of our protein identifications in salivary glands, saliva, and sporozoite
surfaces.

• Line 505: “The fact that the hemolymph immune proteins upregulated
in the proteome of iSGs were not upregulated in the RT-qPCR and in
situ hybridization analysis suggests that these proteins most likely
entered the SGs due to structural damage to the epithelial layer, as
observed in our dextran diffusion assay and previously reported.” The
authors assume that only transcriptional regulation occurred. Could
translational regulation also explain the findings? This hypothesis should
be considered before concluding that leakage from the SGs during
dissection is the sole explanation.

Our *in situ* hybridization analysis confirms that proteins such as
lipophorin and PPO6 are not transcribed in the salivary glands,
supporting the conclusion that these proteins originate from the
hemolymph rather than being transcribed or translated in the salivary
gland. While we cannot entirely rule out the possibility of translational

regulation affecting salivary gland-specific proteins, the majority of
differentially expressed proteins appear to be hemolymph-derived.

• Line 559: “This study offers a functional description of sporozoite
salivary gland interactions...” While the authors have provided an
extensive catalog of proteins involved, the functional relationships
between these proteins remain underexplored. The authors should select
key proteins for further functional studies and justify their choices.

As noted in response to other reviewer comments, to study the effect that
this protein exchange could have in parasite or mosquito biology, we are
developing tools to precisely regulate gene expression in a tissue-
specific manner and at specific stages of parasite development in the
mosquito. While our study provides a comprehensive catalog of proteins
that may influence parasite or mosquito biology, we also highlight key
proteins with functional relationships—such as those involved in
immunity, metabolism, and blood feeding—and discuss their potential
roles in parasite transmission. In addition, we have provided compelling
evidence that sporozoites damage the salivary gland during invasion,
altering the saliva and the structure of the salivary cavities. We believe
these aspects are thoroughly addressed throughout the manuscript.

Additionally, could an AI/ML approach be applied to identify the most
critical proteins for future investigation? How feasible is this method
across different insect species, and could it help pinpoint proteins that
play crucial roles in infection and immune regulation? A discussion on
the potential of such approaches would enrich the manuscript and open
the door for future studies in this area.

Thank you for the suggestion. We agree that AI/ML approaches could
help identify key pathways to target and predict similar interactions
across multiple vector-borne diseases. As discussed above, we are
developing tools to address protein function in follow-up studies. We
have added a discussion of this potential in the conclusion, highlighting
its value for future studies in this area.

Reviewer #4 (Remarks to the Author):

In this manuscript, the authors examined the impact of Plasmodium
infection on Anopheles gambiae salivary glands using proteomics, gene
expression, and morphological analysis and the saliva proteome. The
data provides new insights into the structural and cellular biology for
mosquito salivary glands during sporozoite infection and may contribute
to guiding the development of strategies to interrupt parasite
transmission. The data presented here are very much worthy of

publication. However, there are some details about the methodology and
analysis missing need to be addressed. Below are some comments for
authors to further improve the manuscript.

1. A flowchart of proteomics analyses in fig1 will help to understand
the experimental design.

*We have added a flowchart to Figure 1.*

2. Line 574, in method, only GFP strain was used for proteomics
analysis, but in Supplementary Figure 1: both GFP and mCherry-
infected strains used for proteomics analysis.

*The material and methods section states that: “For the proteome,
we performed infections with a transgenic GFP P. berghei parasite
strain (ANKA GFPcon 259cl2). For all the other assays, we used
P. berghei (Pb) ANKA transgenic line expressing mCherry and
luciferase markers under the control of the hsp70 and eef1a
promoters, respectively.”*

3. All tables are not ready for publication. The tables should be
labeled clearly with table title, column name explanations, et al..
Currently, we have to guess which one is which one.

*A legend has been added for all the Supplementary Tables.*

4. Line 684: The detail method about how the differential expression
(DE) cutoff was decided is not clear.

We have clarified the method for determining differential
expression in the Materials and Methods section. The cutoff for
differential expression was set at a q-value of ≤ 0.05 . The revised
text now reads:

*“Differential expression was assessed using Student’s t-test with*
*permutation-based FDR control. Proteins with q value ≤ 0.05 were*
*considered differentially expressed.”*

In fig 2C, there are some DE proteins have very small fold changes
and very small p value when n=3, it worths to check/explain.

When variance is low, small fold changes with a significant p-
value are common, particularly with an FDR correction, as
consistent measurements across replicates reduce the standard
deviation, leading to statistical significance despite minimal
differences.

5. Line 217: which is sup table 6 and which protein is mitochondrial
pyruvate carrier? The same question for sup table 7?
Protein 12, on table 6 AGAP001417-PB. This table contains
proteins detected only in infected salivary glands. Table 7 contains
proteins detected in both salivary glands, but statistically different.

Legends have been added to the Tables to improve clarity.

6. Line 604: are the 50 mosquitoes the total for all groups or for each
group?

50 mosquitoes were used for each group. We have now clarified this
in the M&M: “*For saliva collection, forced salivation was used on 50*
*female mosquitoes per group.*”

Response to reviewers

Reviewer #1 (Remarks to the Author):

The authors sincerely addressed the questions and concerns raised in my comments and revised the manuscript as appropriate. The additional data strengthened the authors' thesis. I am generally satisfied with the revised manuscript.

Minor comments:

- Line 472: Please correct “oocyts” to “oocyst.”

Addressed

- Supplement Figures 7 and 8: The numbers indicated by “scale bar” in the legends for each figure are “XX”, please describe the correct numbers.

Addressed

- Supplement Figure 7b: A larger image of “sporozoites lodged in the intercellular space” would help readers understand.

Addressed

Reviewer #2 (Remarks to the Author):

The authors have generated new data using transgenic parasite that do not penetrate SG cells. These new data strongly reinforce their hypothesis, rendering the overall manuscript more convincing.

They have also addressed the comments from reviewers. I recommend acceptance.

Reviewer #4 (Remarks to the Author):

The authors have addressed all my questions and my further comment:

Supp figure 1a: need to be polished, both 3 and 4 are “Protein elution/digestion”??

Addressed